# Dynorphin-based "release on demand" gene therapy for drug-resistant temporal lobe epilepsy

Alexandra S Agostinho[1,†], Mario Mietzsch[2,†], Luca Zangrandi[1,‡], Iwona Kmiec[1], Anna Mutti[1], Larissa Kraus[3,4], Pawel Fidzinski[3], Ulf C Schneider[5], Martin Holtkamp[3,4], Regine Heilbronn[2,4,*] & Christoph Schwarzer[1,**]

## Abstract

Focal epilepsy represents one of the most common chronic CNS diseases. The high incidence of drug resistance, devastating comorbidities, and insufficient responsiveness to surgery pose unmet medical challenges. In the quest of novel, disease-modifying treatment strategies of neuropeptides represent promising candidates. Here, we provide the "proof of concept" that gene therapy by adeno-associated virus (AAV) vector transduction of preprodynorphin into the epileptogenic focus of well-accepted mouse and rat models for temporal lobe epilepsy leads to suppression of seizures over months. The debilitating long-term decline of spatial learning and memory is prevented. In human hippocampal slices obtained from epilepsy surgery, dynorphins suppressed seizure-like activity, suggestive of a high potential for clinical translation. AAV-delivered preprodynorphin expression is focally and neuronally restricted and release is dependent on high-frequency stimulation, as it occurs at the onset of seizures. The novel format of "release on demand" dynorphin delivery is viewed as a key to prevent habituation and to minimize the risk of adverse effects, leading to long-term suppression of seizures and of their devastating sequel.

**Keywords** adeno-associated virus; learning; memory; neuropeptide; seizure
**Subject Categories** Genetics, Gene Therapy & Genetic Disease; Neuroscience

## Introduction

With a worldwide prevalence of 1–2%, epilepsy represents one of the most frequent chronic neurological diseases affecting patients of all ages (Thurman *et al*, 2011). Recurrent seizures disrupt normal brain functions, lead to neuronal loss, and result in cognitive and emotional deficits. Patients suffer from stigmatization, social isolation, combined with disability, educational underachievement, and poor employment outcomes (WHO, 2015).

About 70% of epilepsy patients experience focal seizures that arise from an epileptogenic focus in the temporal lobe, most frequently in the hippocampus, called mesial temporal lobe epilepsy (mTLE) (Blumcke *et al*, 2012). Unfortunately, mTLE with hippocampal sclerosis is the hardest to treat with up to 80% of patients not becoming seizure-free with antiepileptic drugs. Moreover, patients suffer from severe adverse side effects (Eadie, 2012; Perucca & Gilliam, 2012). Often, cognitive deficits and emotional blunting develop, potentially facilitated by antiepileptic medication (Ertem *et al*, 2013). In 2008, the FDA issued a black-box warning that several antiepileptic drugs increase the risk of suicidal tendencies (Mula & Sander, 2013). For patients with drug-refractory epilepsy whose seizures originate from a well-defined and accessible focus, neurosurgery for resection of the epileptogenic focus may remain the ultimate solution (Duncan, 2007; Bergey, 2013). But even then, the outcome is highly variable (Spencer & Huh, 2008).

To date, none of the prevailing treatments offers a satisfactory long-term solution. Therefore, medical need for innovative treatment options is high. Accumulating evidence suggests that neuropeptides, including dynorphins, act as endogenous modulators of neuronal excitability (Henriksen *et al*, 1982; Siggins *et al*, 1986). The dynorphins represent a family of vesicle-stored endogenous

1  Department of Pharmacology, Medical University of Innsbruck, Innsbruck, Austria
2  Institute of Virology, Campus Benjamin Franklin, Charité - Universitätsmedizin Berlin, corporate member of Freie Universität Berlin, Humboldt-Universität zu Berlin, and Berlin Institute of Health, Berlin, Germany
3  Department of Neurology, Charité - Universitätsmedizin Berlin, corporate member of Freie Universität Berlin, Humboldt-Universität zu Berlin, and Berlin Institute of Health, Epilepsy-Center Berlin-Brandenburg, Berlin, Germany
4  Berlin Institute of Health (BIH), Berlin, Germany
5  Department of Neurosurgery, Charité - Universitätsmedizin Berlin, corporate member of Freie Universität Berlin, Humboldt-Universität zu Berlin, and Berlin Institute of Health, Berlin, Germany
   *Corresponding author. Tel: +49 30 84453696; E-mail: regine.heilbronn@charite.de
   **Corresponding author. Tel: +43 512 9003 71205; E-mail: schwarzer.christoph@i-med.ac.at
   †These authors contributed equally to this work
   ‡Present address: Institute of Virology, Campus Benjamin Franklin, Charité - Universitätsmedizin Berlin, corporate member of Freie Universität Berlin, Humboldt-Universität zu Berlin, and Berlin Institute of Health, Germany

opioids, perceived as natural anticonvulsants (Tortella & Long, 1988; Mazarati & Wasterlain, 2002). During burst stimulation, typical for the onset of seizures, dynorphins are released from neurons and bind to kappa opioid receptors (KOR), thereby preventing seizure development (Schwarzer, 2009). The seizure threshold is lowered in preprodynorphin (pDyn) knockout mice leading to increased susceptibility for the development of epilepsy (Loacker et al, 2007). Similarly, low dynorphin levels in humans correlate with increased vulnerability for the disease (Stogmann et al, 2002). In mouse models of mTLE as well as in affected patients, dynorphin levels are reduced in the epileptogenic focus, but KOR are mostly maintained (de Lanerolle et al, 1997). The aim of the present study was to replenish the exhausted reservoirs of dynorphins in neurons of the epileptogenic focus. This restores the source of seizure-suppressing endogenous KOR agonists. As vector-derived propeptides are identical to the endogenous protein, they should be similarly stored in large dense-core vesicles, processed, and matured peptides release upon high-frequency stimulation. Thus, transduced dynorphins will be "released on demand" similar to endogenous neuropeptides. In well-recognized animal models of unilateral mTLE, a single focal application of AAV vectors transducing human pDyn into an established epileptogenic focus led to long-term suppression of seizures and stopped disease progression.

# Results

## Suppression of seizures by AAV-pDyn delivery to the epileptogenic focus in the kainic acid mouse model of mTLE

Focal and secondary, generalizing seizures are typical hallmarks of mTLE, as reflected in the widely accepted kainic acid (KA)-induced mouse model of mTLE. Importantly, hippocampal paroxysmal discharges (HPD: unilateral spike trains with a duration of more than 20 s during which animals may display some stereotypies) observed in this model do not respond well to antiepileptic drugs (Riban et al, 2002; Klein et al, 2015; Zangrandi et al, 2016). Therefore, HPDs are considered to model drug-resistant seizures. To study the influence of pDyn overexpression on the frequency and severity of recurrent seizures, an adeno-associated virus (AAV) vector was constructed to express the human preprodynorphin cDNA (Fig 1A).

To achieve high per particle gene expression rates, a self-complementary (sc)AAV2 vector backbone equipped with a potent truncated CBA promoter and translation-enhancing WPRE element was chosen to achieve high-level expression of a codon-optimized human pDyn cDNA, as described in the methods. AAV serotype 1 capsids were chosen for packaging due to proven neuronal transduction combined with restricted spread of the vector beyond the site of injection (Murlidharan et al, 2014; Hocquemiller et al, 2016). Highly purified and concentrated AAV vectors, $2 \times 10^9$ genomic particles (gp) coding for human pDyn (AAV-pDyn), or non-functional control vectors (AAV-ΔGFP) were injected into the epileptogenic focus about 1 month after KA injection, when focal epilepsy had developed. At this stage, animals displayed numerous HPDs (Fig 1B, top EEG trace, and Fig 1C) and up to 3 generalized seizures (displaying spike trains on the traces of all 4 recording electrodes and tonic–clonic motor seizures; Fig EV1) a day. AAV-pDyn delivery induced a gradual reduction in generalized seizures (Fig 1D) and of HPDs (Fig 1E). Generalized seizures completely disappeared within 1 week, and no further events were observed for the entire observation period (3 months). HPDs were gradually reduced over the entire observation period. By contrast, animals injected with AAV-ΔGFP continued to experience seizures for the entire observation period (Fig 1D and E). To demonstrate that Dyn action was mediated by kappa opioid receptors, mice were treated with the KOR antagonist norBNI (20 mg/kg) 30 days post-AAV-pDyn delivery when seizures had disappeared. Drug treatment led to a transient reinstatement of seizures and their disappearance upon washout of the antagonist (Fig 1F).

## Suppression of seizures by AAV-pDyn delivery in an electroconvulsive rat model of mTLE

To confirm that AAV-pDyn-induced seizure suppression in kainic acid-induced mTLE in mice is reproduced in another species and differently induced TLE model, we set up the rat model of electrically induced, focal, and self-sustained status epilepticus (Nissinen et al, 1999, 2000). After unilateral electrical stimulation of the lateral amygdala, rats develop spontaneous seizure-like EEG abnormalities (spike trains) mostly originating from either the amygdala or the hippocampus within a few weeks. To avoid overlap with the stimulation site, AAV-pDyn ($2 \times 10^9$ gp) was infused into the dorsal hippocampus. This treatment induced a significant reduction in spike trains already after 1 week that consolidated up to 4 months (Fig 1G).

## Conservation and restoration of cognitive functions upon AAV-pDyn delivery to the epileptogenic focus

Many mTLE patients suffer from progressive deficits in spatial and declarative memory. Similar comorbidities also develop 1–2 months after unilateral KA injection in mice (Groeticke et al, 2008). To test whether the silencing of seizures in the ipsilateral hippocampus starting during epileptogenesis prevents the memory decline, AAV-pDyn was injected into the epileptogenic focus using mice 2 weeks after KA injection. At this stage, EEG abnormalities such as high-voltage sharp waves, spike trains, and HPD characteristic of early-stage epilepsy have established (Riban et al, 2002), yet animals are still able to learn. Subsequently, mice were tested for spatial memory applying the Barnes maze in a repetitive manner, learning a different target at months 1, 2, and 6 after KA injection. Animals treated with AAV-pDyn performed comparably to naïve, age-matched controls at each time interval (Fig 2, upper panel). Learning curves are depicted in Fig EV2. The decline in performance of both groups of animals at the late time interval (Fig 2G–I) is most probably due to impaired vision known to develop in aged C57BL/6N mice. The time interval chosen is the latest where C57Bl/6N mice are still able to learn the task, and AAV-pDyn-treated epileptic animals kept up with the performance of naïve mice.

By contrast, animals treated with AAV-ΔGFP lost this ability (Fig 2, upper panel). Noteworthy, the animals retained an intact working memory.

All animal groups performed equally well in a spontaneous alternation test. 60 to 75% of correct answers were observed in AAV-pDyn- or AAV-ΔGFP-treated epileptic and in naïve animals (Fig EV4K). This finding recapitulates data from mTLE patients, in

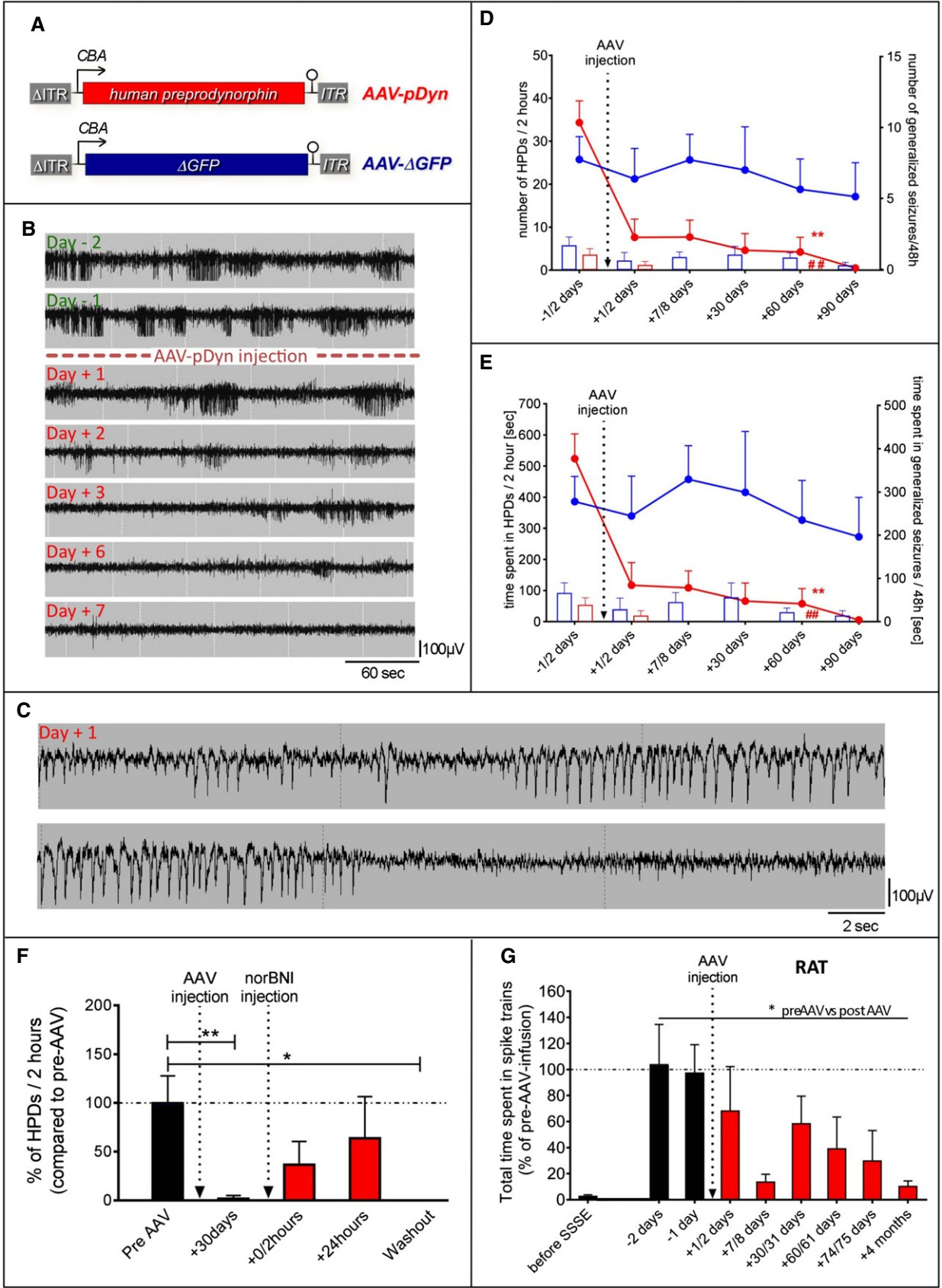

Figure 1.

**Figure 1. Effect of AAV-pDyn on seizures in mouse and rat models of TLE.**

A   Displayed (sc)AAV2-based vector backbones were packaged in AAV serotype 1 capsids. AAV-ITRs are displayed in gray, and ΔITR refers to the mutated ITR version of scAAVs. The transgene of AAV-pDyn is a codon-optimized version of the full-length human preprodynorphin cDNA enhanced by a WPRE element. Control vectors carry either a truncated, non-functional version of the enhanced GFP gene (AAV-ΔGFP), or its functional counterpart (AAV-eGFP; not displayed).

B   Kainic acid mouse model of TLE: Daily EEG recordings obtained from the epileptogenic focus starting from 1 month after KA injection and spanning the period from 2 days before to 7 days after AAV-pDyn delivery ($2 \times 10^9$ gp).

C   Higher time resolution of the indicated section of the Day +1 in (B), representing a hippocampal paroxysmal discharge (HPD). A generalized seizure is depicted in Fig EV1.

D, E   The characteristic EEG features of this model, secondary generalized seizures (bars) and HPDs (lines), were reduced in number and in duration by AAV-pDyn (red; $n = 3$ (from day 60); 7 (till day 30) per time interval), but not by AAV-ΔGFP or after sham treatment (blue; $n = 4$ (day 90); 5 (day 30 and 60); and 6 (before day 30), per time interval). The relatively high variability of seizure frequencies and duration is typical for this model and reflects the findings in human mTLE. Some animals could not be recorded for the entire period due to loss of implant. $**P < 0.01$; effect of treatment on HPDs; $^{##}P < 0.01$; effect of treatment on generalized seizures; analyzed by two-way ANOVA with Bonferroni correction for both number and time.

F   Injection of norBNI (20 mg/kg; i.p.) results in a transient reappearance of HPDs immediately and 24 h after application. One week after norBNI application (washout), suppression of HPDs was re-established. Data obtained from 4 epileptic animals before (black) and after (red) AAV-pDyn delivery ($2 \times 10^9$ gp) are depicted. $*P < 0.05$; $**P < 0.01$; one-way ANOVA for repeated measures with Friedman post hoc test.

G   EEG recordings obtained from the ipsilateral dorsal hippocampus of rats after electrical self-sustained status epilepticus (SSSE) before (black) and after (red) AAV-pDyn delivery ($4 \times 10^9$ gp) are depicted. Spike trains with a frequency of at least 2.5 Hz induced by SSSE were markedly reduced by AAV-pDyn ($n = 4$). $*P < 0.05$ one-way ANOVA for repeated measures with Friedman post hoc test.

Data information: Data represent mean ± standard error of the mean.

whom the short-term memory regularly remains intact (Silva *et al*, 2010).

In a second experiment, mice in the state of chronic epilepsy, already displaying established deficits in spatial learning and memory, were tested for functional reconstitution. Although the contralateral hippocampus displays only minor neuropathological alterations, its function appears compromised. Due to strong collateral connections between both hippocampi, disturbances of the non-affected, functionally intact, contralateral hippocampus are highly likely.

This led to our hypothesis that silencing of the ipsilateral hippocampus by pDyn delivery should prevent this development. Mice were treated by AAV delivery into the epileptogenic focus 5 weeks after KA injection. Animals injected with AAV-pDyn gradually regained lost spatial memory within 1 month after AAV delivery. Performance levels similar to naïve controls were achieved within 2 months (Fig 2J–L). By contrast, no improvement of memory functions was observed in epileptic mice treated with AAV-ΔGFP (Fig 2M–O). The time course of reconstitution fits well to the observed time course of seizure suppression in mice observed before (Fig 1).

## Neuron-specific long-term expression of pDyn and stimulation-dependent "release on demand"

Long-term expression of AAV-delivered pDyn in neurons of the epileptogenic focus was demonstrated at 6 months after KA

injection (5.5 months after vector delivery) by double immunofluorescence. Mostly dentate granule cells and pyramidal cells showed cytoplasmatic pDyn immunoreactivity together with nuclear immunoreactivity for the neuronal marker NeuN, but not for the glial marker GFAP (Fig 3A–F). Expression of Dyn peptides was also observed in principle and non-principal neurons. Upon injection of AAV-pDyn into the hilus of naïve mice, strong immunoreactivity was observed in the mossy fiber tract, indicative of transduction of granule cells. Moreover, somata of non-principle cells in the polymorph cell layer and corresponding labeling in the outer molecular layer suggests transduction of GABAergic interneurons. Strong labeling of the inner molecular layer also in the contralateral hippocampus suggests transduction of mossy cells (Fig EV3).

It is well established that pDyn is stored and processed to mature peptides in large dense-core vesicles (for review, see Schwarzer, 2009). Indeed, we found that AAV-pDyn transduction increased the levels of Dyn peptides in the dorsal hippocampus of epileptic animals at 1.5 months after AAV-pDyn delivery, as quantified by an EIA specific for mature Dyn B. Overall Dyn levels and the observed interanimal variability decreased 6 months after application (Fig 3G). A transient, early increase in gene (Dyn) expression is typical for high copy numbers of delivered AAV vectors. Only a fraction of the initially transduced AAV genomes persist long-term as nuclear episomes (Murlidharan *et al*, 2014; Hocquemiller *et al*, 2016). Importantly, the early increase in neuronal Dyn did not lead to increased Dyn levels in the cerebrospinal fluid (CSF) at

**Figure 2. Effects of AAV-pDyn on spatial learning and memory.**

A–I   Spatial learning and memory were tested on the Barnes maze. Quadrant 1 (Q1) contains the target hole (red; A). Unilateral injection of AAV-pDyn (B) or AAV-eGFP (C) into naïve young adult mice (12 weeks age) did not influence the performance as compared to naïve controls (D) when tested 4 weeks after AAV injection. Mice treated 2 weeks after KA with AAV-pDyn (E, H) performed equally to age-matched naïve controls (D, G) 2 weeks (E) and 5.5 months (H) after treatment. By contrast, animals treated 2 weeks after KA with AAV-ΔGFP (F, I) gradually lost this ability. Two-way ANOVA revealed significance between AAV-ΔGFP- and AAV-pDyn-treated groups for interaction 2 weeks ($P = 0.0349$) and 5.5 months ($P = 0.0311$) after AAV, respectively, at each time interval and quadrant ($P < 0.0001$) 2 weeks after AAV. Comparison of AAV-pDyn injected with naïve animals revealed no differences.

J–O   Epileptic mice, which were not able to learn the Barnes maze task 1 month after KA (J, M), AAV-pDyn application restored spatial memory 1 (K) and 2 months (L) after treatment. Treatment with AAV-ΔGFP did not result in improved memory (N, O). Two-way ANOVA revealed significance for interaction ($P = 0.0049$) and quadrant ($< 0.0001$) comparing AAV-ΔGFP with AAV-pDyn-treated animals at the later time interval.

Data information: Data represent mean ± standard deviation. Animal numbers: (B) and (C) $n = 9$; (D) through (H) $n = 8$; (I) $n = 5$, note: Three mice had to be killed due to accelerating seizure activity and resulting weight loss; (J) through (O) $n = 7$. $***P < 0.001$; $**P < 0.01$; $*P < 0.05$ by one-way ANOVA and Dunnett post hoc test.

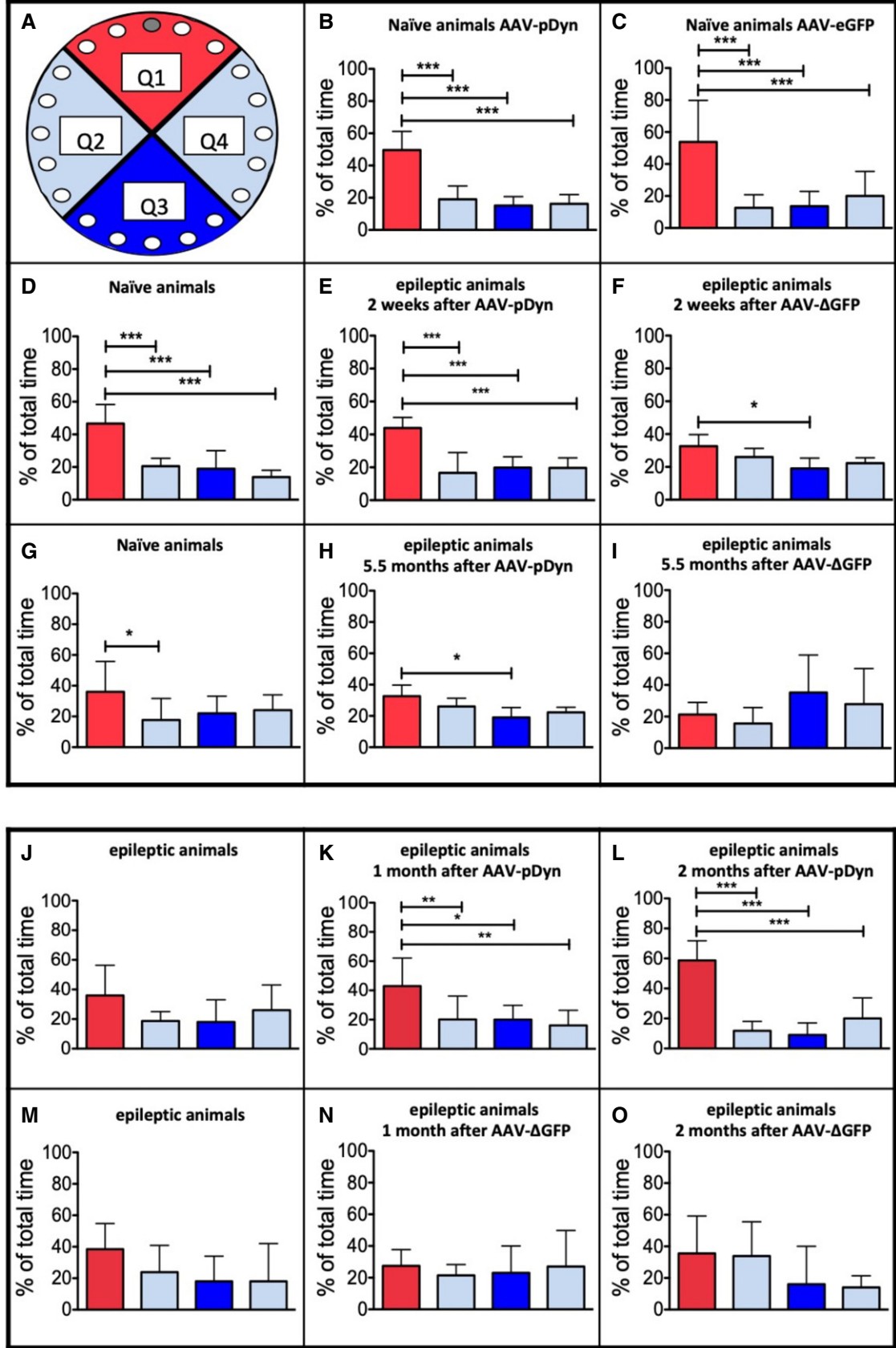

Figure 2.

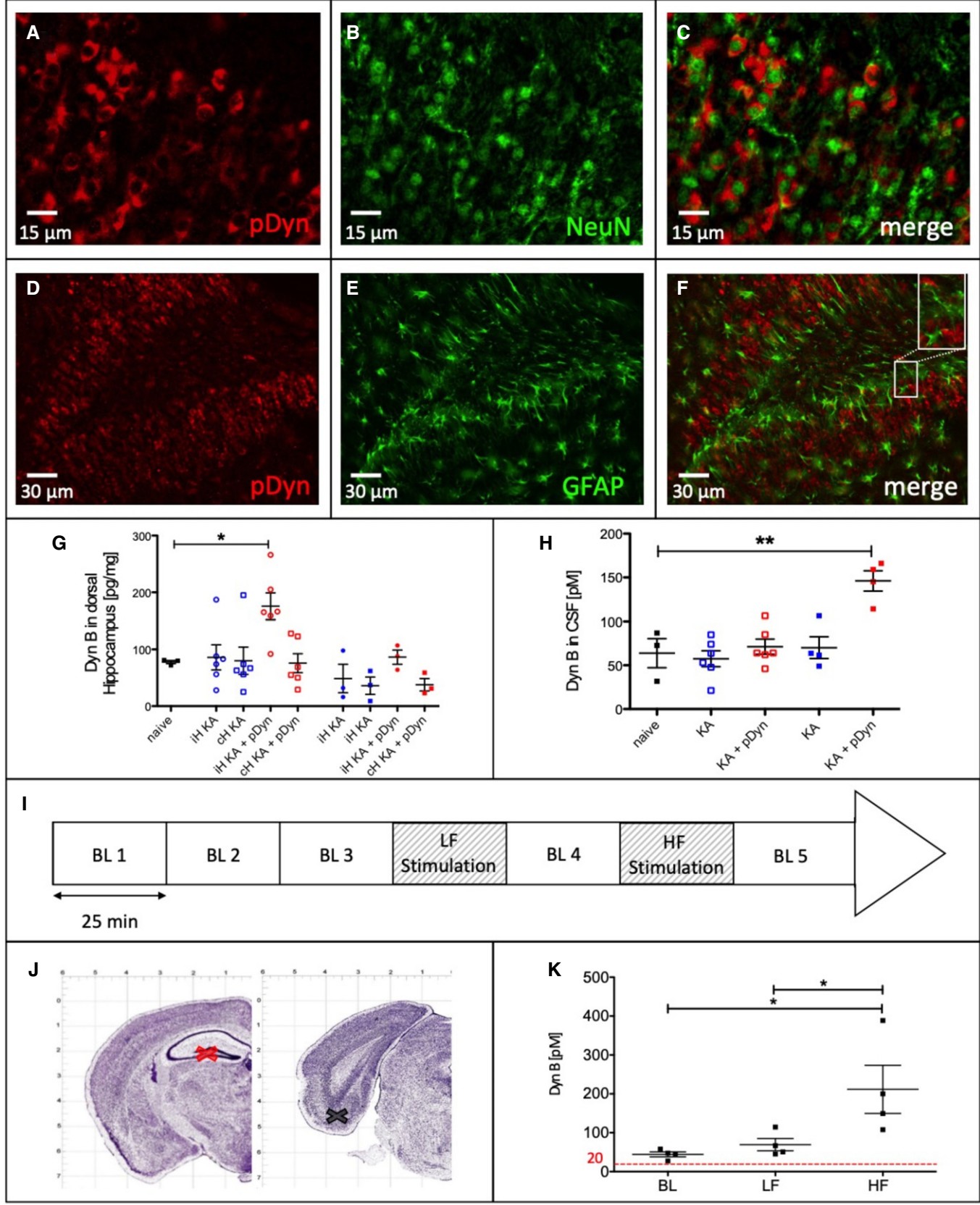

**Figure 3.**

**Figure 3.  Distribution and release "on demand" of vector-derived dynorphins.**

A–F  Double-immunofluorescence labeling is depicted for pDyn and NeuN (A–C) or GFAP (D–F) in the ipsilateral dentate gyrus of KA-treated and AAV-pDyn-injected mice. Enlarged view in (F) represents 15 × 30 μm.

G  Mature Dyn B content (measured by a Dyn B-specific EIA) in the dorsal hippocampus of mice treated with KA (blue symbols) or KA and AAV-pDyn (red symbols) 1.5 (open symbols; n = 6) and 6 (filled symbols; n = 3) months after vector treatment. Naïve animals were age-matched to the 1.5 months after AAV group. iH stands for ipsilateral hippocampus and cH for contralateral hippocampus. *P < 0.05; paired t-test was used for comparison of ipsi- and contralateral hippocampi. Two-way ANOVA was used to compare Dyn levels between the early and late time interval.

H  Mature Dyn B content in the CSF of mice treated with KA (blue symbols) or KA and AAV-pDyn (red symbols) 1.5 (open symbols; n = 6) and 7 (filled symbols; n = 4) months after vector treatment. **P < 0.01; one-way ANOVA with Dunnett post hoc test

I  The release of fully processed, mature Dyn B under different stimulation conditions was analyzed in microdialysates collected from the hippocampus of pDyn-deficient (KO) animals 2 weeks after injection of AAV-pDyn. Three baseline samples (BL; 25 min each) were collected. This was followed by 25 min low-frequency stimulation (LF), 25 min baseline, and 25 min high-frequency stimulation (HF; I).

J  The microdialysis probe (red cross) was placed in the dentate gyrus, and the stimulation electrode (black cross) in the entorhinal cortex.

K  Dyn B was quantified by EIA in the dialysate collected during different stimulation intensities. The red line represents the detection limit of the EIA. *P < 0.05; n = 4; one-way ANOVA with Tukey post hoc test.

Data information: Data represent mean ± standard error of the mean.
Source data are available online for this figure.

1.5 months after AAV-pDyn transduction (Fig 3H). Interestingly, a twofold increase in Dyn was observed in the CSF at 6 months after AAV-pDyn transduction. This most probably reflects increased release. In any case, the amount of Dyn in the CSF still represents non-efficacious concentrations in the pM range (Fig 3H). For KOR activation, about 100-fold higher Dyn concentrations were reported as EC50 in the established GTPγS assay (Joshi et al, 2017).

To investigate whether AAV-pDyn-derived peptides are mature and released in a stimulation-dependent manner, AAV-pDyn was injected into the hippocampus of naïve pDyn knockout animals. After 2 weeks, animals were subjected to perforant path stimulation, and microdialysis of released peptides was performed (Fig 3I–K, lower panel). The release of dynorphin B derived from AAV-pDyn was quantified by an ELISA specific for mature dynorphin B, but insensitive to precursors. Upon low-frequency stimulation, dynorphin levels in the extracellular fluid were close to the baseline. Only high-frequency stimulation resulted in a significant, sixfold increase in dynorphin release (Fig 3I). These findings indicate that AAV-pDyn-derived propeptides are processed—at least in part—to maturity and released "on demand", i.e., high-frequency stimulation, as is the case at the onset of seizures.

### Dyn peptides suppress seizure-like activity in diseased human hippocampi

The above experiments had established that mature Dyn peptides are released from transduced AAV-pDyn upon high-frequency stimulation. The decisive step toward clinical translation of AAV-pDyn as gene therapy vector is to demonstrate that Dyn is similarly active in human disease. To evaluate anticonvulsant effects of Dyn in human epileptic tissue, we measured field potentials in area CA1 of human hippocampi obtained from epilepsy surgery. Epileptiform activity was induced by elevating KCl to 8 mM and by application of 100 μM 4-AP. After at least 20 min of stable baseline seizure activity (Fig 4A), Dyn A and Dyn B (Dyn A/B) were bath-applied (600 nM, respectively) for at least 20 min (Fig 4B). During washout, either Dyn A/B was removed from the solution (patient 1+2), or the KOR antagonist 5′-GNTI was co-applied (patient 3+4; Fig 4C; 150 nM). Burst events (Fig 4D) and interictal spikes (Fig 4E) were observed in the CA1 region under elevated KCl conditions. As shown in Fig 4F, the amplitude and number of burst events are markedly reduced.

Consequently, the interevent interval is markedly prolonged upon application of Dyn A/B in three out of four specimens from individual patients (Table EV1). The effect is reversed under washout conditions, or after co-application of the KOR antagonist 5′-GNTI except for patient 1, where the quality of the recordings faded during the washout period. Notably, the reappearance of burst-like activity upon washout under co-application of a strong KOR antagonist (5′-GNTI) demonstrates both, the reversibility of burst repression, and the specificity of Dyn action via kappa opioid receptors. For details on single patient's samples, see Table EV1. As expected, and as known from mTLE mouse models interictal spikes remain largely unaffected (Fig 4G). However, their role in epileptogenesis is still unclear. Removal of spike-generating tissue upon epilepsy surgery is considered non-predictive for a favorable outcome (Jacobs et al, 2010). In addition, interictal spikes are no indication for epileptogenicity (Usui et al, 2008). It is hypothesized that interictal spikes result from abnormal firing within an otherwise normal network, in contrast to fast ripples (bursts) which probably reflect epileptogenic reorganization (van Diessen et al, 2013).

### Safety of AAV-pDyn in the hippocampus of naive mice

To explore the safety of focal AAV-pDyn delivery, naïve animals were injected with AAV-pDyn or AAV-eGFP into the left hippocampus. A battery of behavioral tests was performed from 2 to 9 weeks after AAV delivery. Anxiety and stress-coping ability, as well as working memory performance, were comparable among all groups tested (Fig EV4), suggesting no influence of pDyn overexpression. In brain, slices obtained 10 weeks after AAV-pDyn delivery focally restricted, yet strong pDyn mRNA expression was detected, covering large parts of the dorsal hippocampus (Fig 5A). Importantly, no indication of cell toxicity or inflammation was seen in the brains 10 weeks after AAV-pDyn or AAV-eGFP (Fig 5B–F).

## Discussion

In this report, we present "proof of concept" for a novel therapeutic concept, a neuropeptide-based "release on demand" gene therapy to control seizures in focal epilepsy. We show for the first time that AAV-mediated delivery of pDyn to the epileptogenic focus of established TLE leads to virtually complete and long-lasting suppression

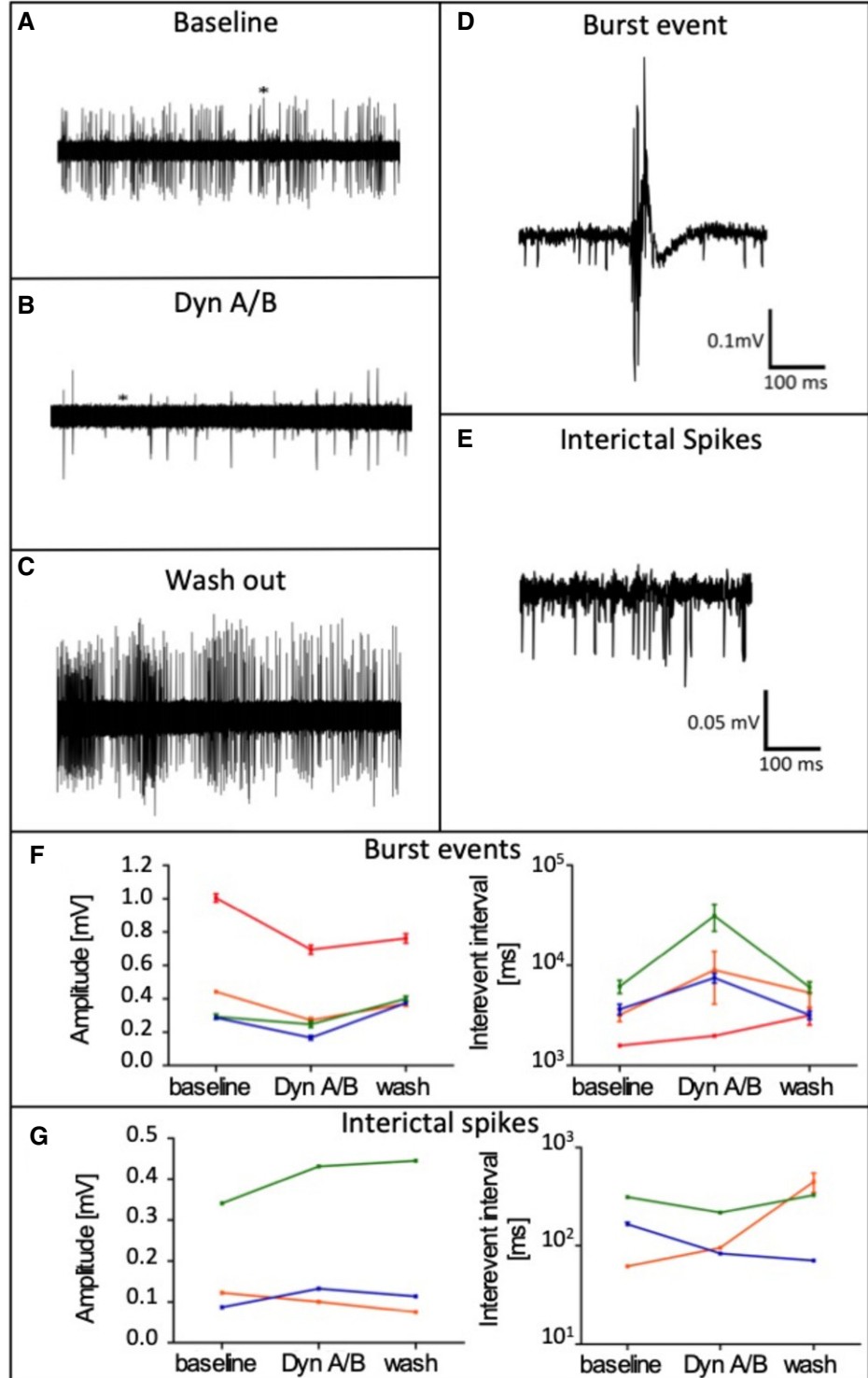

**Figure 4. Effect of Dyn A/B on induced epileptiform activity in human hippocampal slices.**

A–C  Representative traces under conditions of increased KCl (A), together with Dyn A/B (B), or under co-application of Dyn A/B with 5′-GNTI (C). * In (A, B) indicate the time interval shown in (D, E).

D, E  Burst events (D) and interictal spikes (E) were recorded from area CA1.

F, G  Treatment with Dyn A/B markedly reduced the amplitude and number of bursts (F) but not of interictal spikes (G). Each color represents the last 5 min of each application for one slice of 4 individual patients. Data sets for each patient represent mean ± SEM for amplitude or absolute number of events. Two-way ANOVA for repeated measurements revealed high significance not only for interaction ($P < 0.0001$) and amplitude ($P < 0.0001$) or interevent intervals ($P < 0.0001$), but also for the factor patient ($P < 0.0001$). Therefore, careful interpretation of the raw data is considered as preferred method. For detailed data, see Table EV1.

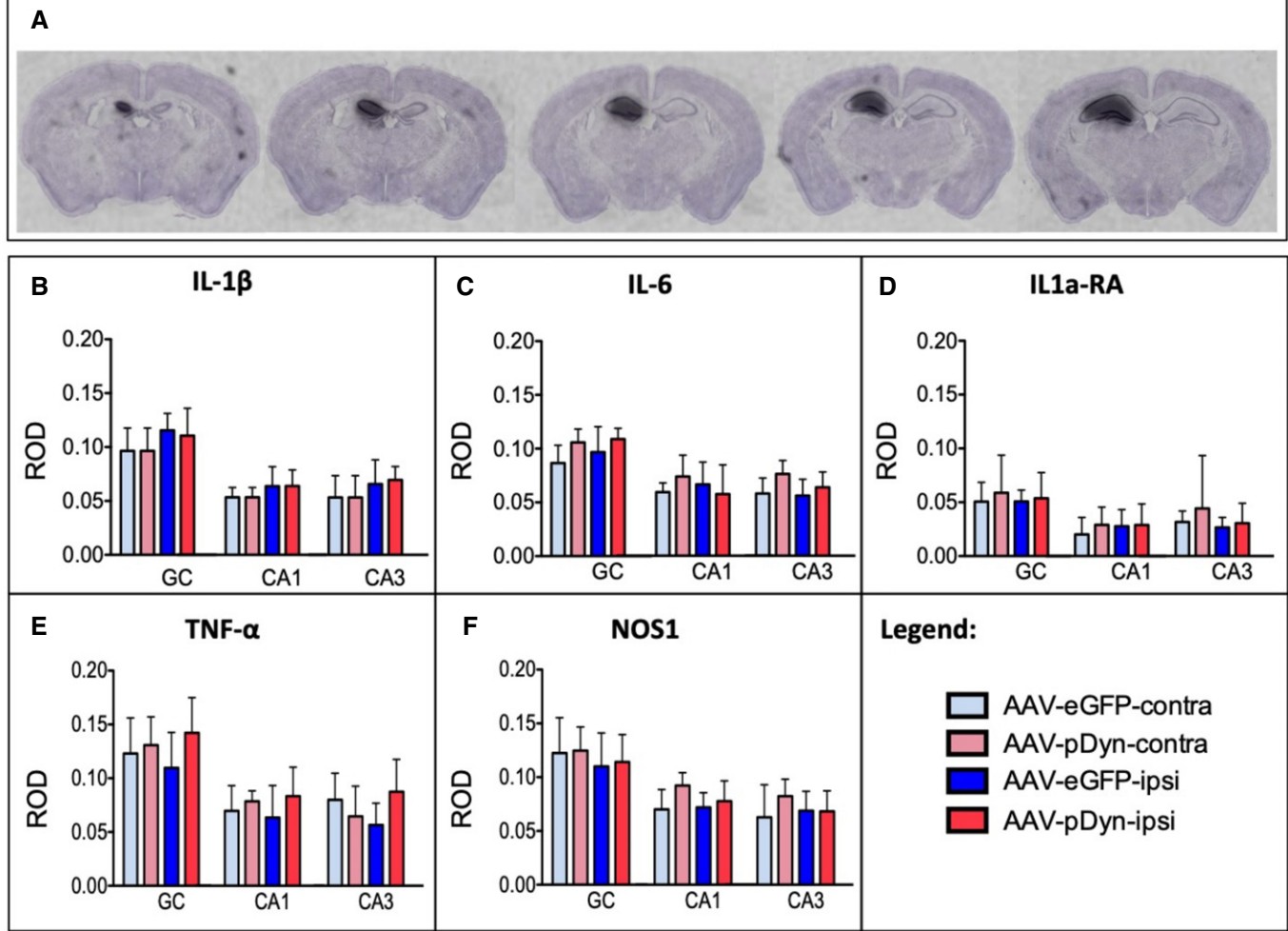

**Figure 5. Vector spread and analysis of inflammatory effects of AAV-pDyn.**

A   In situ hybridization for AAV-derived pDyn mRNA at 10 weeks post-AAV-pDyn delivery in the left dorsal hippocampus is depicted for a single brain at five levels between 1.1 (left) and 2.3 (right) mm from bregma. The probe specifically detects the codon-optimized sequence; therefore, the contralateral side represents a control devoid of the respective mRNA.

B–F   At 10 weeks post-AAV-pDyn delivery, markers of inflammation were analyzed as follows: Relative optical densities (ROD) were evaluated from film autoradiographs after *in situ* hybridization for mRNAs of inflammatory markers (interleukin-1β (B), interleukin-6 (C), interleukin-1a receptor agonist (D), tumor necrosis factor-α (E), and nitric oxide synthase-1 (F)) in 3 principal sub-regions of the hippocampus in AAV-pDyn ($n = 7$; red)- or AAV-eGFP ($n = 5$; blue)-injected animals. GC = granule cell layer; CA = cornu ammonis. Data represent mean ± standard deviation.

of focal and generalized seizures over months. This is also the first demonstration that a one-time gene therapy can restore the TLE-typical decline of learning and memory.

Dynorphins represent a family of endogenous opioids that modulate neuronal excitability through activation of KOR. KOR activation by drug agonists has long been known to suppress seizures (Simonato & Romualdi, 1996). Dynorphins are therefore viewed as endogenous anticonvulsants.

In epileptic tissue, dynorphin levels are reduced, both in TLE patients (Hurd, 1996; de Lanerolle *et al*, 1997) and in mouse models of TLE (Simonato & Romualdi, 1996; Schwarzer, 2009), thereby facilitating neuronal over-excitability. KOR are largely maintained and remain responsive to agonists (de Lanerolle *et al*, 1997). Reduced dynorphin levels combined with intact KOR in TLE led to

the concept that focal delivery of pDyn could restore its neuronal supply and suppress seizures. Highly expressed neuropeptides stored in large dense-core vesicles had been shown to serve as reservoirs of peptide drugs (Clynen *et al*, 2014). Here, we show that AAV-delivered pDyn yields mature peptides, which are released in a stimulation-dependent manner. Triggered by high-frequency stimulation at the onset of seizures, the focally restricted reservoir of releasable, protective neuropeptides is delivered "on demand" and dampens the local network. As a consequence, AAV-derived dynorphins prevent imminent seizures and preserve neuronal functions without being permanently secreted (Fig 6).

The mouse model of KA injection into the dorsal hippocampus closely mimics the characteristics of mesial temporal lobe epilepsy (mTLE) encountered in patients. HPDs observed in this model do

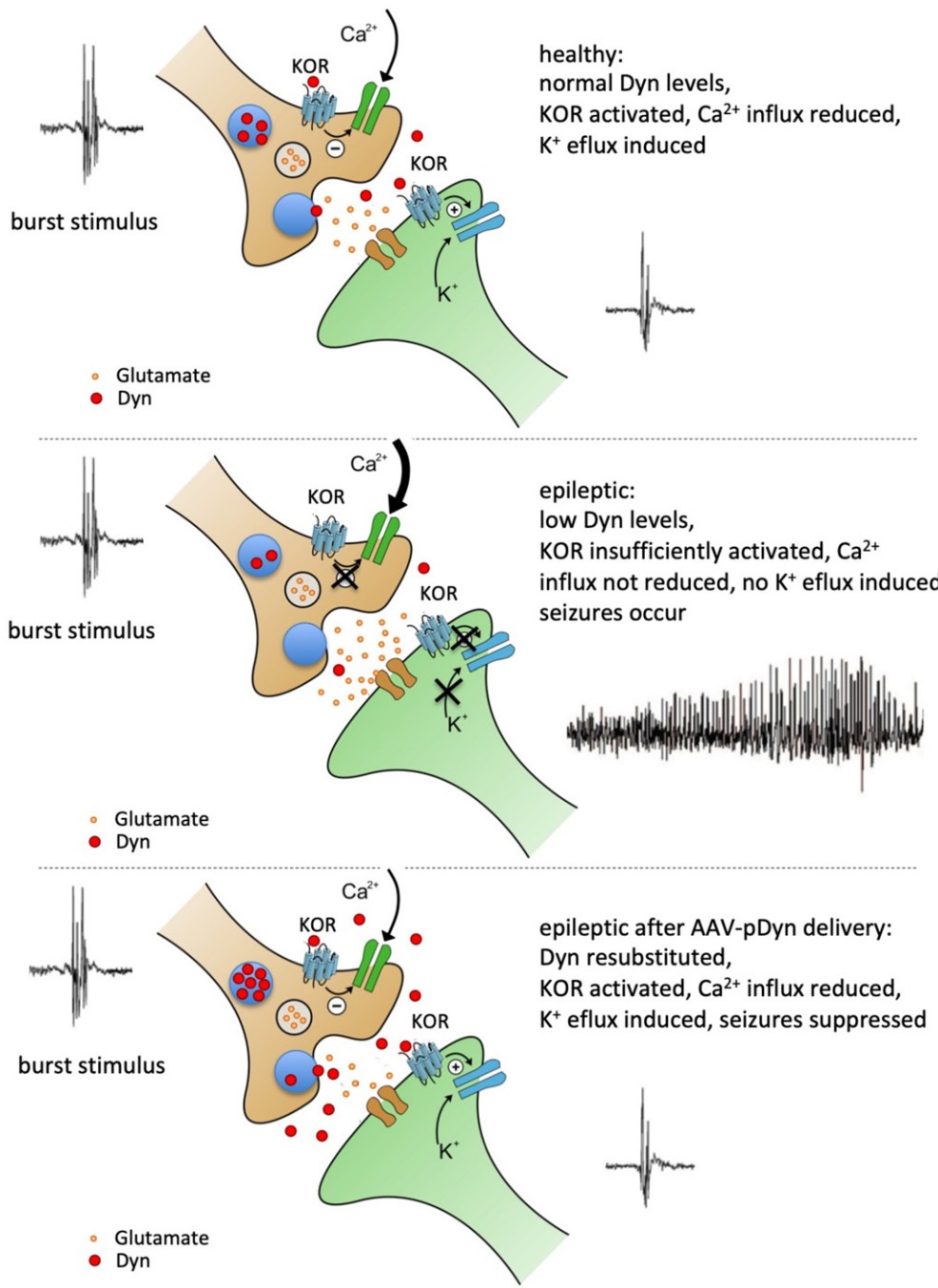

**Figure 6. Modulation of synaptic transmission by dynorphin.**

Graphical summary of the alterations in the dynorphinergic system between healthy (top panel) and epileptic (middle panel) hippocampi and their consequences on neuronal signaling. The bottom panel indicates the proposed mode of action of AAV-pDyn gene therapy. The incoming burst stimulus triggers the release of dynorphins in a "release on demand" mode.

not fully respond to antiepileptic drugs (Riban *et al*, 2002; Klein *et al*, 2015; Zangrandi *et al*, 2016) and thus resemble drug-resistant seizures in mTLE patients. Another hallmark of mTLE is secondary, generalizing seizures. We demonstrate that both mTLE hallmarks were largely suppressed by pDyn overexpression. Activation of presynaptic KOR decreases N-, L-, and P/Q-type $Ca^{2+}$ currents

(Rusin *et al*, 1997), leading to a reduction in glutamate release. Postsynaptic KOR activation stimulates voltage-gated $K^+$ channels (Moore *et al*, 1994; Madamba *et al*, 1999), resulting in hyperpolarization and dampening of neuronal excitability. We have recently shown that non-peptidergic KOR agonists suppress HPDs in epileptic mice to a higher extent than modern antiepileptic drugs for other

targets (Zangrandi *et al*, 2016). The findings presented here imply that AAV-pDyn-derived peptides exert similar, yet lasting effects. Suppression of seizures at the site of origin was accompanied by long-term conservation of spatial memory over the entire observation period of 6 months, while control animals lost this ability between 1 and 2 months after induction of epileptogenesis. Hippocampi are strongly interconnected (Gloor *et al*, 1993), and epileptic discharges spread to the contralateral hippocampus. This induces functional disturbances also in the initially intact, contralateral side. Our data suggest that the extensive and lasting suppression of seizures in the epileptogenic hippocampus conserves (during epileptogenesis) or rehabilitates (in chronic epilepsy) the functions of the contralateral hippocampus. Importantly, both, seizure suppression and rehabilitation of brain functions, were achieved in animals with fully established epilepsy, characterized by recurrent focal seizures, hippocampal sclerosis, and granule cell dispersion, thus in a stage with overt clinical symptoms.

Although the mouse model of KA injection into the dorsal hippocampus is widely accepted and applied to study antiepileptic drug action, the translatability of the findings can hardly be judged from a single model. Therefore, the ability of AAV-pDyn to reduce seizures in the rat model of electrically induced status epilepticus provides independent evidence for its effectivity. Furthermore, the ability of Dyn peptides to reduce induced burst charges in slices obtained from human epileptic hippocampi removed during focus surgery proves that high levels of mature Dyn peptides are effective to dampen focal seizure development in human epileptic tissue. The latter experiment is as close as one can get to the patient, before testing AAV-pDyn in a clinical trial.

AAV vectors have long been proven as efficient and safe for long-term gene transduction in the CNS (Heilbronn & Weger, 2010; Murlidharan *et al*, 2014), backed by data from a series of clinical studies (Hocquemiller *et al*, 2016). AAV serotype 1 leads to efficient but strictly focal, neuronal transduction. A single dose of $2 \times 10^9$ AAV vector genomes (gp) per animal was sufficient for long-term suppression of hippocampal HDPs and of generalized seizures. The presented preliminary studies on the safety profile of AAV-pDyn are promising: So far, no signs of toxicity, inflammation, or immune reactions were recorded. No adverse effects were observed in healthy mice treated with AAV-pDyn, suggesting that unilateral hippocampal overexpression of pDyn does not interfere with emotional control or stress-coping abilities, effects attributed to KOR activation in distinct brain regions (Bilkei-Gorzo *et al*, 2008; Wittmann *et al*, 2009).

Gene therapy for focal epilepsy has been a target for focal delivery of galanin or neuropeptide Y (NPY) alone or together with its receptor. While NPY shows activity on human hippocampus slices obtained from TLE patients, galanin was not (Ledri *et al*, 2015). Numbers and duration of seizures were reduced, but long-term suppression of seizures in a therapeutic setting is still outstanding (Noe *et al*, 2010) (Ledri *et al*, 2016). A glutamate-sensitive chloride channel was tested only in acute seizures (Lieb *et al*, 2018). Recently, a mutated voltage-gated potassium channel Kv1.1 proofed beneficial in the model of systemic KA injection (Snowball *et al*, 2019).

Gene therapy for focal epilepsy with AAV-pDyn represents a novel therapeutic option to overcome drug resistance and induce lasting seizure suppression. AAV-pDyn may be delivered before, simultaneously, or after surgery. Instead, focal gene therapy with AAV-pDyn offers a less invasive treatment option to silence the epileptic focus. AAV-pDyn impedes neither drug treatment nor resection of the focus. Moreover, AAV-pDyn offers the prospect to prevent or even reverse the imminent downhill process of cognitive and/or emotional decline. The expectation that AAV-pDyn delivery to a chronically diseased hippocampus may rescue the contralateral hippocampus from progressive disease is very encouraging for patients with long-standing intractable disease. Considering the high medical need for a definite and long-term solution, the "release on demand" concept has enormous advantages over permanent and systemic drug treatment. Timely and spatially restricted peptide release from a permanently refilled reservoir limits the development of tolerance and reduces off-target adverse effects, by taking the brain "off-treatment" in interictal phases while guaranteeing availability at the onset of seizures. Last but not least, it will be fascinating to see the "peptide on demand" concept being transferred to other targets and disease states, and its realization by focused AAV delivery.

## Materials and Methods

### Animals

Male adult C57BL/6N wild-type and pDyn knockout (pDyn-KO) mice (Loacker *et al*, 2007) were investigated in this study. Mice were group-housed (max. 5 mice in type IIL cages) with free access to food and water. Male adult Sprague Dawley rats (Charles River Lab., Sulzberg, Germany) were used for the self-sustained status epilepticus model. Rats were housed in pairs in type III cages with free access to food and water. Temperature was fixed at 23°C and 60% humidity with a 12-h light–dark cycle (lights on 7 am to 7 pm). All procedures involving animals were approved by the "Austrian Animal Experimentation Ethics Board" in compliance with the "European Convention for the Protection of Vertebrate Animals Used for Experimental and Other Scientific Purposes" ETS no.: 123. Every effort was taken to minimize the number of animals used. The study was designed in compliance with the ARRIVE guidelines.

### Drugs

Kainic acid was purchased from Ocean Produce International, Sandy Point, Canada. *nor*-Binaltorphimine (norBNI) and 5′-Guanidinyl-17-(cyclopropylmethyl)-6,7-dehydro-4,5α-epoxy-3,14-dihydroxy-6,7-2′,3′-indolomorphinan (5′-GNTI) were purchased from Bio-Techne Ltd, Abingdon, UK. Drugs were dissolved in phosphate-buffered saline.

### Kainic acid injection and electrode implantation

Male mice (12–15 weeks) were injected 20 min prior to surgery with the analgesic meloxicam (2 mg/kg). Animals were sedated with ketamine (160 mg/kg, i.p.; Graeub Veterinary Products, Switzerland) and subsequently deeply anesthetized with sevoflurane through a precise vaporizer (Midmark, USA). Mice were injected with 1 nmol KA in 50 nl solution into the hippocampus (RC −1.80 mm; ML +1.80 mm; DV −1.60 mm with the bregma as a reference point) as previously described (Zangrandi *et al*, 2016). Electrodes were implanted

immediately after KA administration as described elsewhere (Zangrandi *et al*, 2016). For electrodes, see Appendix Fig S1.

### Self-Sustained Status Epilepticus (SSSE) model: electrode implantation

Male rats (10–12 weeks) were injected 20 min prior to surgery with the analgesic meloxicam (2 mg/kg). Animals were sedated with ketamine (80 mg/kg, i.p.; Graeub Veterinary Products, Switzerland) and subsequently deeply anesthetized with sevoflurane through a precise vaporizer (Midmark, USA). For amygdala stimulation, rats were implanted with a bipolar electrode (diameter 0.127 mm, DV distance between the tips 0.4 mm; Franco Corradi, Milano, Italy) into the lateral nucleus of the left amygdala (Nissinen *et al*, 2000); coordinates of the lower electrode tip, according to the rat brain atlas of Paxinos and Watson (2007) (RC −3.30 mm; ML +5.50 mm; DV −8.2 mm with bregma as reference point). Bilateral depth electrodes were implanted into the hippocampi (RC −4.00 mm; ML ± 2.60 mm; DV −3.6 mm with bregma as reference point) as previously described (Zangrandi *et al*, 2016); a cortical electrode was positioned on the right motor cortex, and a reference/ground screw electrode was placed over the cerebellum as described for mice (Zangrandi *et al*, 2016).

### Self-Sustained Status Epilepticus (SSSE) model: amygdala stimulation

Two weeks after the surgery, a 1-h baseline EEG was recorded from each rat. SSSE was induced by electrically stimulating the lateral nucleus of the left amygdala. The protocol for the stimulation followed precisely the published setup (Nissinen *et al*, 2000). For the stimulations, a fixed voltage power supply (Voltcraft FSP-1132), a stimulus isolation unit (ISO-01D, NPI), and a Master-8 pulse stimulator (A.M.P.I.) were used. After 3–4 weeks, EEGs were recorded from amygdala, ipsi- and contralateral hippocampi, and cortex to verify the presence of secondary generalized seizures and other focal EEG abnormalities induced by the electrical stimulation.

### AAV constructs

Self-complementary (sc) AAV2-ITR-flanked vector backbones were constructed with the CBA promoter, consisting of the HCMV-IE gene enhancer followed by the chicken beta-actin gene promoter. Expression of the full-length, codon-optimized, human pDyn cDNA enhanced by the woodchuck hepatitis virus post-transcriptional regulatory element (WPRE) or of an inactivated variant of EGFP is followed by the bovine growth hormone poly A signal sequence (Fig 1A).

### AAV vector preparation

AAV vectors were produced by two-plasmid (pDG1-rs) cotransfection in HEK 293 cells essentially as described (Mietzsch *et al*, 2014). AAV vector backbones were packaged in serotype 1 capsids and HPLC-purified from benzonase-treated, cleared freeze–thaw supernatants by one-step AVB sepharose affinity chromatography using 1 ml prepacked HiTrap columns on an ÄKTA purifier (GE Healthcare). Peak fractions were dialyzed against PBS and the titers of the highly purified AAV preparations were determined by qPCR,

measured as AAV DNA-containing genomic particles (gp) per ml, as described before (Mietzsch *et al*, 2014).

### AAV injections

For experiments with subsequent AAV administration, a guide cannula was implanted, which was attached to the hippocampal depth electrode and targeting the hilus of the dentate gyrus. For AAV injections, animals were mildly anesthetized in a sevoflurane chamber during the time of the injection (20 min). The injection was made through the guide cannula with an injection pump at a flow of 0.1 µl/min, and a total volume of 2 µl was injected. For rats, a total volume of 4 µl was injected in each hippocampus (both ipsi- and contralateral) with a pump flow of 0.2 µl/min.

### EEG recording and analysis:

The EEG was obtained using a wireless recording device (Neurologger, TSE, Germany) and automatically analyzed using SciWorks Software (DataWave Technologies, USA). EEGs were filtered for epileptiform spikes defined as a high-amplitude discharges (> 3 × baseline) lasting < 70 ms. Spike trains were defined as the occurrence of at least three spikes with a frequency higher than 1 Hz and lasting for at least 1 s. Spikes with lower frequencies were counted as interictal spikes. In mice, prolonged hippocampal paroxysmal discharges (HPD) were defined as spike trains lasting for a minimum of 10 s (Zangrandi *et al*, 2016). In rats, hippocampal focal abnormalities induced by the electrical stimulation were defined as spike trains lasting for a minimum of 2 s and frequency higher than 2.5 Hz. Generalized seizures were assessed as co-appearance of high-voltage EEG abnormalities in both, hippocampal depth and motor-cortical surface electrode. The HPDs were evaluated during a period of 2 h; hippocampal focal abnormalities in rats were evaluated during a period of 3 h. Generalized seizures were evaluated for a minimal period of 48 h.

### Behavioral testing

Mice were transferred to the anteroom of the testing facility 24 h before the commencement of experiments. Tests were video-monitored and evaluated by an experimenter blinded to the treatment of the animals.

To assess learning and memory of naïve and treated animals, the Barnes maze test was executed at 60 lux on a flat circular table (diameter 100 cm) with 20 holes around its perimeter. Only one allowed the mouse to exit the maze into an escape dark box. The position of the escape box was kept constant during the entire experiment. Visual clues were placed around the disk with an interval of 90°. On the first day, mice were allowed to freely explore the maze during 5 min with the target hole open. Acquisition was made during the next 4 days. Three trials of 3 min maximum were performed. After the mouse found the hole, the box was closed and the animal was kept in there for 2 min to let it associate the escape box as a secure place. If the animal did not find the target hole during the maximum time given, it was gently guided to the hole. On the sixth day, all holes were closed and the mouse was free to explore the maze during 5 min. For evaluation, the board was divided into quadrants and the time spent in each was measured. The quadrant containing the previously open

escape hole is referred to as Q1, which is flanked by Q2 and Q4, while Q3 is opposing Q1.

The procedures for the open-field, elevated plus maze and light–dark choice tests were performed as recently published (Wittmann *et al*, 2009) in accordance with the recommendations of EMPRESS (European Mouse Phenotyping Resource of Standardized Screens; http://empress.har.mrc.ac.uk).

In short: The open-field arenas had a size of 50 × 50 cm and were illuminated to 150 lux. Mice were observed over 10 min, measuring time, distance travelled, and number of entries into three subfields: center (central 16% of overall area), border (8 cm along walls), and intermediate.

The light–dark test was performed in the same arenas with a black box inserted, which covered 1/3 of the area. Light was set to 400 lux and mice tested for 5 min measuring time, distance travelled, and number of entries into the light compartment.

The elevated plus maze test consisted of 4 arms, two closed (20 cm walls) and two open arms, each 50 × 5 cm in size elevated about 80 cm aboveground. Exploratory activity on the open arms was tested over 5 min at 180 lux.

The forced swim test was performed in a single 15-min trial at a water temperature of 25°C. Immobility, defined as no activity for at least 2 s, was independently evaluated from video clips for the interval from the second to the sixth minute and for the final 4 min by two investigators blinded to the genotype and/or treatment of the animals.

The tail suspension test was performed in a single 6-min trial. Mice were elevated about 20 cm above a plate light to 150 lux. All tests were performed between 9 a.m. and 1 p.m.

The spontaneous alternation test was conducted in a symmetrical "Y" shaped maze, at 50 lux during 8 min. An alternation was defined as a triplet of sequential unique location visits. The alternation score was calculated by dividing the number of correct alternations by the total number of alternations.

## Microdialysis

Microdialysis was performed on pDyn-KO animals, which had received AAV-pDyn injection into the left hippocampus as described above 2 weeks before (RC −2.0 mm; ML +1.1 mm; DV −2.0 mm). At the time of viral vector injection, animals were implanted with a guide cannula targeting the hilus of the injected hippocampus and a stimulation electrode in the entorhinal cortex (RC −4.20 mm; ML +3.20 mm; DV −4.90 mm). For microdialysis, MAB-2 probes (cutoff 35 kDalton; 2-mm exposed tip; SciPro, Sanborn, NY) were placed into the hippocampus and flushed by artificial CSF containing (in mM): NaCl (140); KCl (3.0); $CaCl_2$ (1.25); $MgCl_2$ (1.0); $Na_2HPO_4$ (1.2); $NaH_2PO_4$ (0.3); and glucose (3), pH 7.2 at a rate of 0.4 µl/min. aCSF baseline was collected for 3 × 25 min without stimulation, followed by 25 min low-frequency stimulation (150 µA; isolated 0.3-ms square pulses at 0.1 Hz, ISO-STIM 01D, NPI, Tamm, Germany). After another 25-min baseline free of stimulation, 25 min of high-frequency stimulation was performed (150 µA; 1-s. trains of 0.3-ms square pulses of 50 Hz; trains were separated by 10 s.) followed by a 25-min baseline.

## Dynorphin B enzyme immunoassay (EIA)

CSF samples were collected from deeply anesthetized mice as described elsewhere (Liu & Duff, 2008). Dorsal hippocampi were dissected and immediately frozen on dry ice. Neuropeptides were extracted using 1 M acetic acid as described earlier (Merg *et al*, 2006). The content of dynorphin B in the dialysate of microdialysis experiments, CSF, and hippocampal extracts was measured by EIA (S-1429; Peninsula; San Carlos, CA) according to manufacturer's manual. In short, samples were incubated with the antiserum for 1 h, followed by an overnight incubation with tracer. On the second day, streptavidin-horse radish peroxidase was added for 1 h after five washes with EIA buffer. After another five washes, samples were reacted with TMB solution for 5 min and then analyzed on a plate reader a 450 nm. Dyn B content was analyzed based on calibration samples run in parallel and expressed as ng/ml. The quantity of dynorphin B present in the dialysate was measured by reading absorbance values at 450 nm with the Perkin Elmer Wallace 1420 Victor 2 microplate reader.

## Electrophysiology on human hippocampal slices

Experiments were approved by the Ethics Committee of Charité—Universitätsmedizin Berlin on 01.11.2014 (EA2/111/14) and are in agreement with the Declaration of Helsinki. All patients gave their written consent prior to the surgery.

Specimens were collected, transported, and processed in cold carbogenated NMDG-aCSF (95% $O_2$, 5% $CO_2$) containing (in mM): NMDG (93), KCl (2.5), $NaH_2PO_4$ (1.2), $NaHCO_3$ (30), $MgSO_4$ (10), $CaCl_2$ (0.5), HEPES (20), glucose (25), Na-L-ascorbate (5), thiourea (2), and Na-pyruvate (3). 400-µm slices were cut using a vibratome (Leica VT1200S) within 40 min. from removal and stored in an interface chamber. During a recovery period of at least 5 h, slices were continuously perfused with carbogenated aCSF containing (in mM): NaCl (129), $NaH_2PO_4$ (1.25), $CaCl_2$ (1.6), KCl (3.0), $MgSO_4$ (1.8), $NaHCO_3$ (21), and glucose (10) (1.6 ml/min, 32°C).

## Recordings

For recordings, slices of patients 1 and 2 were transferred to a standard submerged chamber and perfused at 4 ml/min. Slices of patients 3 and 4 were recorded in a modified version of the membrane chamber (Hill & Greenfield, 2011) and perfused at roughly 10 ml/min. In both conditions, recordings were performed at 32°C.

Field potential recordings were performed with borosilicate pipettes (1.5 mm outer diameter, Science Products) pulled with a vertical puller (Narishige, PC-10) (1–2 mΩ) and filled with NaCl (154 mM). Electrodes were placed in the pyramidal cell layer of CA1. Signals were low-pass-filtered at 2 kHz, sampled at 10 kHz by a Digidata 1550 interface, and processed by PClamp10 software (Molecular Devices, Sunnyvale, CA, USA).

Epileptiform activity was induced by elevating KCl concentration to 8 mM and application of 100 µM 4-AP. After at least 20 min of stable baseline activity, dynorphin A and dynorphin B (Dyn A/B) were bath-applied (600 nM, respectively, Bachem) for at least 20 min. During washout, either Dyn A/B was removed from the solution or the κ opioid receptor antagonist 5′GNTI was co-applied (150 nM, Tocris). Recordings were analyzed with Clampfit 10 threshold analysis to measure the event frequency and amplitude. All events judged as burst events (with positive deflection and duration > 400 ms) or interictal spikes (no positive deflection, duration < 200 ms) were manually indicated for further analysis in Clampfit. To eliminate contamination

### The paper explained

#### Problem

Focal epilepsy is one of the most common chronic CNS diseases where seizures regularly arise from circumscribed foci in the brain. The most common clinical subform is so-called temporal lobe epilepsy (TLE) where the focus lies in the hippocampus or the adjacent amygdala, key sites for learning, memory, and emotional control. Unfortunately, up to 80% of affected patients do not respond sufficiently to antiepileptic drugs. Invasive brain surgery to remove the focus is available for preselected cases, but even then seizure freedom is not guaranteed. Unfortunately, ongoing seizures may lead to sclerosis of the hippocampus and a gradual decline of learning capabilities, memory, and emotional control. The resulting cognitive and mood effects are often much more devastating for patients and families than the seizures themselves. Medical need is high to find better therapeutic options.

#### Results

We have developed a novel AAV vector-based gene therapy for focal delivery of protective neuropeptides, called dynorphins. Using well-recognized rodent models for temporal lobe epilepsy, the vector was injected into the established epileptic focus when seizures had developed. Seizure frequency went down within days, and seizures disappeared by 2 months and stayed off thereafter. Learning and memory were fully maintained, even restored in models of chronic epilepsy. Also in human, epileptic tissue dynorphins repress seizure-like burst activity. AAV-transduced dynorphins are expressed and stored in neurons restricted to the focus and the release is dependent on high-frequency stimulation, as is the case at the onset of seizures. This novel "release on demand" format for induced release of therapeutic peptides is viewed as the key for the observed long-term suppression of seizures without signs of tolerance.

#### Impact

With long-term data from two therapeutic animal models and strong *ex vivo* evidence for dynorphin activity in diseased epileptic tissue from individual patients, the translational impact is high for the novel AAV gene therapy. Dynorphin is delivered by a minimal-invasive one-time application of a small vector dose to the affected site in the brain. The chosen AAV vector capsids' clinical safety profiles and documented long-term focal gene expression of other genes in the CNS of primates and human patients give high hopes for the development of a long-term solution for patients suffering from drug-resistant temporal lobe epilepsy.

of burst event amplitude by interictal spikes, events were low-pass-filtered at 100 Hz to measure absolute amplitude.

## Immunohistochemistry

For immunohistochemical analysis, mice were transcardially perfused with 4% PFA in PBS. Brains were cut to 40-µm coronal sections on a vibratome and stored in PBS with 0.1% sodium azide at 4°C. Brain sections were blocked in blocking buffer (50 mM Tris–HCl, 150 mM NaCl, 0.2% Triton X-100 (SERVA Electrophoresis GmbH), 10% NGS (normal goat serum Vector Laboratories), 0.1% sodium azide) for 1.5 h at room temperature. Thereafter, brain slices were incubated overnight at room temperature in blocking buffer containing primary antibodies: pDyn (Neuromics—400217) together with antibodies directed against either NeuN (Millipore—LV151948) or GFAP (ACRIS—1224001). After washing slices three times for 10 min in washing buffer (50 mM Tris–HCl, 150 mM NaCl, 0.2% Triton X-100), brain slices were incubated in washing buffer containing fluorescence-labeled secondary antibodies (Cy3: Jackson ImmunoResearch—66095; Alexa Fluor 488: Invitrogen Molecular Probes—66095). Slices were washed in PBS and mounted in Vectashield medium (Vector Laboratories).

## *In situ* hybridization

Mice were killed by cervical dislocation, and brains were quickly removed and snap-frozen in −50°C 2-methylbutane. Brains were subsequently sliced into 20-µm sections using a cryostat. *In situ* hybridization was performed as described in detail elsewhere (Wittmann *et al*, 2009). In brief, six different $^{35}$S-labeled single-stranded 45mer antisense DNA-oligonucleotides complementary to human codon-optimized pDyn (5′-GGT ACT TCC GCA GGA AGC CGC CGT ATC TCT TCA CTT GTT CTT TGG-3′); inducible nitric oxide synthase (5′-GAT GTG CTG AAA CAT TTC CTG TGC TGT GCT ACA GTT CCG AGC GTC-3′); TNF alpha (5′-TAC AGA CTG GGG GCT CTG AGG AGT AGA CAA TAA AGG GGT CAG AGT-3′); interleukin-1b (5′-TAC TGC CTG CCT GAA GCT CTT GTT GAT GTG CTG CTG CGA GAT TTG-3′); interleukin-6 (5′-GCC ACT CCT TCT GTG ACT CCA GCT TAT CTG TTA GGA GAG CAT TGG-3′); and IL-1 receptor antagonist (5′-GTT GGT GAT TAT TAC AGG CCT CGG CAG TAC TAT TGG TCT TCC TGG-3′) mRNAs were used. Coronal sections were incubated with radiolabeled oligonucleotides for 18 h at 52°C. Brain slices were washed 4 × 15 min in 1× SSC (saline sodium citrate buffer) at 62°C, dried, and exposed to MR film (Kodak BioMax) for 8–14 days. Relative optical densities were analyzed from film autoradiographs using ImageJ 64 software.

## Statistical analysis

Prism 5 for Mac was used for statistical analysis. Two-way ANOVA with Bonferroni correction was used for comparison of treatment groups in EEG analysis and behavior and Dyn levels in hippocampi at different time intervals after AAV-pDyn application. One-way ANOVA with Newman–Keuls multiple comparison test was applied to analyze quadrant preference in the Barnes maze test and microdialysis experiments. Paired *t*-test with manual correction for multiple testing was used to compare Dyn levels in ipsi- and contralateral hippocampi. A *P* value < 0.05 was considered significant.

Data are presented as mean ± standard error of the mean (SEM) for electrophysiology on human slices, *in vivo* EEG on rodents, and EIA analysis and mean ± standard deviation (SD) for all other experiments.

Experimenters were unaware of the treatment of animals while analyzing data. For behavioral experiments, animals were distributed in a pseudo-randomized way, splitting litters into different groups. For EEG studies, animals were allocated to viral vector treatment in a way to ensure equal distribution of animals with high and low numbers of seizures in both groups. For behavioral testing, animals were distributed in a pseudo-randomized way, splitting litters into different groups. In experiments where vectors were given after initial testing, good and bad learners were equally distributed between groups.

**Expanded View** for this article is available online.

## Acknowledgements

We want to thank Stefan Weger, and Daniela Hüser, Charité Univer-sitätsmedizin Berlin, for advice on AAV vector production; Gerald Obermair, Medical University of Innsbruck, for statistical advice; Asla Pitkänen, University of Eastern Finland, for advice on the rat epilepsy model and for provid-ing the bipolar electrodes; and Inge Kapeller for excellent technical support. This work was supported by the Austrian Science Fund (FWF; I-977, P-30592, and W-1206) to C.S., the German Research Foundation (DFG He1598/9-1), the SPARK program of BIH and Stiftung Charité, Berlin (INV_PRO_452), to R.H.

## Author contributions

ASA performed most *in vivo* analysis in mice, early interval behavioral studies, EEG recordings, and EIA and immunohistochemistry; RH designed the AAV-pDyn vector; MM constructed and generated all AAV vectors; LZ developed the EEG analysis and recording, and the microdialysis experiment; IK performed late interval Barnes maze experiments; AM performed and analyzed the rat experiments, analyzed mouse EEG, and prepared figures; and CS assisted by a technician performed *in situ* hybridizations and peptide analysis. UCS performed epilepsy surgery and provided patient samples; LK and PF performed electrophysiology on human hippocampi, and together with MH analyzed the human data. CS and RH conceived and supervised the entire study and conceptualized, drafted, and finalized the manuscript with equal contributions.

## Conflict of interest

A PCT application of the described technology is pending. R.H. is inventor of a patent related to rAAV technology and owns equity in a company commercial-izing AAV for gene therapy.

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
