## [Review Process File · EMBO Molecular Medicine]

Dynorphin-Based “Release on Demand” Gene Therapy for Drug-Resistant Temporal Lobe Epilepsy

Alexandra S. Agostinho, Mario Mietzsch, Luca Zangrandi, Iwona Kmiec, Anna Mutti, Larissa Kraus, Pawel Fidzinski, Ulf C. Schneider, Martin Holtkamp, Regine Heilbronn, Christoph Schwarzer.

Review timeline:

Submission date:	20 th October 2018
Editorial Decision:	17 th December 2018
Revision received:	27 th May 2019
Editorial Decision:	29 th July 2019
Revision received:	1 st August 2019
Accept:	6 th August 2019

Editor: Celine Carret

Transaction Report:

1st Editorial Decision

17th December 2018

Thank you for the submission of your manuscript to EMBO Molecular Medicine and please accept our apologies for the delay in getting back to you. We have now heard back from the three referees whom we asked to evaluate your manuscript.

You will see from the comments pasted below that all referees find the data interesting and promising, however several items need to be critically addressed before publication. Of particular importance we would insist on explaining the discrepancies noted by referee 3 in the experimental procedure. Referee 1 is mainly concerned about long-term effects and would like to see what's happening 3-4 months down the line (not 6 months); safety of the approach should also be better documented (the mechanism of action in the epilepsy context and the lack of toxic effect related to the pathway involved in epilepsy and eventually others, should be studied). Finally some rewriting and provision of controls and better figures are needed.

We would welcome the submission of a revised version within 3 to 4 months for further consideration and would like to encourage you to address all the criticisms raised as suggested to improve conclusiveness and clarity.

REFeree REPORTS

Referee #1 (Comments on Novelty/Model System for Author):

strong unmet medical need.
well characterized model used
large panel of technics

Referee #1 (Remarks for Author):

The paper by Alexandra S. Agostinho et al , Dynorphin-Based "Drug on Demand" Gene Therapy for Drug-Resistant Temporal Lobe Epilepsy describes a gene therapy approach for a severe form of local epilepsy, called mesial temporal lobe epilepsy (mTLE) . This mTLE with hippocampal sclerosis is the hardest to treat and is associated with cognitive deficits.

The dynorphins are a family of endogenous opioids, modulators of neuronal excitability. they are implicated in addiction. They are perceived as natural anticonvulsants.

During burst stimulation, typical for the onset of seizures, dynorphins are released from neurons and bind to kappa opioid receptors (KOR), preventing seizure development

Preprodynorphin (pDyn) knockout mice show increased susceptibility for the development of epilepsy. Dynorphin levels in humans correlate with increased vulnerability for the disease. In mouse models of mTLE as well as in affected patients, dynorphin levels are reduced in the epileptogenic focus, while KOR are mostly maintained. Altogether, these results suggest that delivering Dy in affected regions could be of therapeutic value.

The aim of the present study was to replenish the exhausted reservoirs of dynorphins in neurons of the epileptogenic focus. Dynorphin, stored in vesicles can be released and activate KOR to suppress seizures.

The paper is well written but more explanations could be directly put in the figures to help reading;

Authors show nice and documented results in terms of prevention of seizures evidently demonstrated by this gene delivery strategy. The use of two different well-recognized animal models in mouse and rat is convincing in the short term. However there are several important concerns that should be addressed to consider the paper for publication

Major comments

1) long-term results

Results on spatial learning and memory are not convincing in the long-term (5,5 months) (Figure 2). This is puzzling. This result and the decreased long term production shown in Figure 3G argue for a decreased expression / efficacy in the long-term. This should be more clearly analyzed and discussed. Escape in the long-term ? Long term production of pDyn should be evaluated. This is also contrasting with CSF measures (see my comment below discussing results presented page 9)

2) Safety of the approach in naive mice must be addressed more precisely (particularly histology) and in the long-term.

Overexpressing Dyn, even in a restricted portion of the brain could have consequences

Evaluation was made up to 10 weeks after vector administration. This is not sufficient Long term study is needed.

Cognitive consequences should be evaluated in bilaterally injected animals to increase the sensitivity of the test.

Histology should be much more detailed with long-term (at least 6 months) time points and detailed analysis : neuronal aspect (picnosis, neuronal markers), neuronal count , inflammation (microglia, astrocytes) with high magnification images. Comparison to non-injected animals or contralateral hemisphere should be made. Readouts of Dyn action should be analysed : KOR signaling in particular

3) This release of the dynorphin is called by authors "on demand". This term leads to misinterpretation and should be changed.

Transgenic Dyn is properly matured and behaves like endogenous counterpart and delivered following stimulation. However the term on demand is confusing and not appropriate. The production of the transgenic dyn is not controlled. Overexpression of the pDyn is permanent. Only the release of mature Dyn vesicles is upon stimulation. This notion should be more clearly explained and it is the opinion of the reviewer that the term « on-demand » is confusing, could be misinterpreted, and should not be used particularly in the title. Since one issue with gene therapy is that gene expression is not regulated and transduced gene is permanently expressed, this is of importance.

Even the release of the mature form stored in vesicles seems to rely on stimulation, transgenic

Dynorphin is produced constitutively under the CAG promoter which raises questions :
 What is the level of production of transgenic Dyn compared to endogenous protein. Total Dyn protein should be quantified.
 Does it accumulate in cells ? in what compartment ? (high quality histology should be performed to document that) How is it eliminated ?
 Authors should investigate the mechanism of action, signal that target correct maturation of pDyn and down signalling mechanism on KOR : show readout of KOR signaling

Since Dynorphins have important functions in the brain, particularly involving reward processing, mood control and the development of addiction, these issues are of major importance in the perspective of a potential therapeutic use.

Specific comments

Injection protocol:

It would help adding a scheme showing the technics and implanted material. This would help understanding the following Methods: For experiments with subsequent AAV administration, a guide cannula was implanted, which was attached to the hippocampal depth electrode and targeting the hilus of the dentate gyrus.

At the time of viral vector injection, animals were implanted with a guide cannula targeting the hilus of the injected hippocampus and a stimulation electrode in the entorhinal cortex (RC -4.20 mm; ML +3.20 mm; DV -4.90 mm). For

Figure 1:

The cassette presented in a single strand cassette and not a self complementary cassette. This should be modified. The serotype 2 UTR are used and the capsid is 1 . This should be added.

Legend of Figure 1:

It is mentioned that control vectors carry either a truncated, nonfunctional version of the enhanced GFP gene (AAV-ΔGFP), or its functional counterpart (AAV-eGFP; not displayed). Why is eGFP mentioned here ? Where is it used ?

Figure 3: How is measured DynB content in G

page 9:

It is stated that « Dyn expression decreases 6 months after application (Fig 3G) and that the early increase of neuronal Dyn did not lead to increased Dyn levels in the cerebrospinal fluid (CSF) at 1.5 months after AAV-pDyn transduction (Fig. 3H). Interestingly a 2-fold increase of Dyn was observed in the CSF at 6 months after AAV pDyn transduction »

This seems in contradiction With 3G results. Moreover this raises the issue of potentially elevated Dyn level in the CSF. In the context of a supposed limited expression in a restricted region of the brain, how this can be interpreted?

Page 11:

To investigate whether AAV-pDyn derived peptides Dyn peptides suppress seizures-like activity in human hippocampus . How is this experimental design and the amount of Dyn delivered are relevant to the AAV-pDyn delivery? (in the micodyalisis exp the levels are 200 pm).

Graphic summary: should integrate overexpressing Dyn condition in healthy cells

Sup figure 2: images and histological analysis should be strongly improved.

Discussion

It is said in the discussion that « A single dose of 108 AAV vector genomes (gp) per animal was sufficient for long-term suppression of hippocampal HDPs and of generalized seizures. The safety profile of AAV-pDyn is excellent: no signs of toxicity, inflammation or immune reactions were recorded. No adverse effects were observed in healthy mice treated with AAVpDyn, suggesting that unilateral hippocampal overexpression of pDyn does not interfere with emotional control or stress coping abilities, effects attributed to KOR activation in distinct brain regions » This must be moderated with regards to my comments both on the limited tox informations (in particular

histology) and on the short term study only performed.

A comment should be made on other gene therapy approaches for this disease and how AAV-Dyn competes with that.

The conclusion stating that « considering the high medical need for a definite and long-term solution the "drug on demand" concept has enormous advantages over permanent and systemic drug treatment. » is misleading. The « drug » is the AAV-pDYN that could be potentially injected to the patient. The product would be indeed permanently delivered and produced.

This conclusion and the sentence below should be strongly moderated, not to oversell the work. « Timely and spatially restricted peptide release limits the development of tolerance and reduces off-target adverse effects, by taking the brain "offtreatment" in inter-ictal phases. Last but not least, it will be fascinating to see the "peptide on demand" concept being transferred to other targets and disease states, and its realization by focused AAV delivery. »

Again the interesting result of this work is that not only pDyn is produced locally but it is correctly matured and delivered upon epileptic stimulation leading to beneficial effect.

Referee #2 (Comments on Novelty/Model System for Author):

The authors employed appropriate model systems on all levels.

Referee #2 (Remarks for Author):

The authors present a range of studies that establish the anti-seizure efficacy and safety of viral vector based preprodynorphin transduction and expression. The studies were well designed in order to address many of the issues inherent to anti-seizure gene therapy. Essentially the diverse findings provided a convincing "proof of principle".

The one minor comment regards the quality of panels A thru C in figure 3. The NeuN immunofluorescence quality is quite low and there are clearly pDYN positive cells that do not co-localize with NeuN. Do the authors have a better example?

Referee #3 (Comments on Novelty/Model System for Author):

This is an interesting paper supporting the idea that viral vector delivery of inhibitory neuropeptides to an epileptic focus could be an effective gene therapy approach for drug-resistant focal epilepsy. They use appropriate rodent models, however a weakness of this paper in my opinion is that their reduction in seizures and epileptiform activity is reported for a relatively small number of animals and during short recording sessions.

Referee #3 (Remarks for Author):

Agostinho and colleagues present an interesting paper expanding preclinical data studies that indicate viral vector driven over expression of inhibitory neuropeptides can have therapeutic efficiency in rodent models of temporal lobe epilepsy.

'On demand' release of an anti-seizure peptide has several advantages, namely that it is released only during high frequency neuronal firing. This therefore reduces potential concerns of homeostatic compensation in response to permanently expressed and functional proteins as used in some other gene therapy approaches.

The authors demonstrate that this treatment can also improve memory function in previously epileptic rodents.

Below are my comments on the manuscript presented.

1. Although Fig 5a shows that there is a good spread of transduced cells in the dorsal hippocampus

can the author's state what the viral transduction volume is? Is expression restricted just to the dorsal hippocampus or does it also transduce the ventral hippocampus. 108 viral particles seems quite low, many studies report much higher titre 1012/1013.

2. How long does it take for virally transduced cells to make the protein of interest? Many groups wait 1-3 weeks after AAV transduction to detect sufficient expression of the transgene protein. However in Fig 1B there appears a significant reduction in epileptiform activity and seizures within 1-2 days.

3. The recording period of EEG is very short. 2hrs for HPD analysis and 48hrs for detecting seizures. Seizure frequency in particular tends to be non-normally distributed. Seizures cluster therefore in a 48hr window seizure frequency could be skewed. A longer baseline record period (7-14 days) and longer post treatment recording sessions (preferably continuous) would be more convincing.

4. Fig 1C shows a blow up of an HPD, can they also provide a blow up of a seizure.

5. Fig 1D - why are the n numbers changing during the course of the experiment - (n=3-7).

6. Fig 1F The colour of the preAAV in AAVDyn rats (blue) is a little confusing as the colour blue is used in other panels to denote AAV-GFP.

7. What is the half-life of norBNI? How long would you expect it to have an effect?

8. If there is a release in suppression of HPDs within 0-2hrs post-injection of norBNI does this indicate that dynorphin is constitutively being released?

9. Fig 1G - Control AAV-GFP data is missing from this figure.

10. Is the dose of kainic acid correct? Pg 18 line 24, '50nl of 1nM KA', most people use 50-100nl of 20mM KA. Also there is a formatting error here, reference 16.

11. 4 electrodes were implanted but analysis only performed on the electrode in the ipsilateral hippocampus? With time there is a development of a more dispersed epileptic network, does dynorphin injection prevent the development of abnormal epileptiform discharges in other brain areas?

12. Does dynorphin overexpression reduce neuropathology? Is there a reduction in sclerosis, granule layer dispersion relative to animals injected with AAV-GFP?

13. Is the conclusion that the recovery in memory impairment is due to reduced seizure and HPDs alone or also due to improvement or prevention of worsening in neuropathology?

Point by point responses to the reviewers' comments:

Reviewer 1:

Referee #1 (Remarks for Author):

The paper by Alexandra S. Agostinho et al , Dynorphin-Based "Drug on Demand" Gene Therapy for Drug-Resistant Temporal Lobe Epilepsy describes a gene therapy approach for a severe form of local epilepsy, called mesial temporal lobe epilepsy (mTLE) . This mTLE with hippocampal sclerosis is the hardest to treat and is associated with cognitive deficits.

The dynorphins are a family of endogenous opioids, modulators of neuronal excitability. they are implicated in addiction. They are perceived as natural anticonvulsants.

During burst stimulation, typical for the onset of seizures, dynorphins are released from neurons and bind to kappa opioid receptors (KOR), preventing seizure development

Preprodynorphin (pDyn) knockout mice show increased susceptibility for the development of epilepsy. Dynorphin levels in humans correlate with increased vulnerability for the disease. In mouse models of mTLE as well as in affected patients, dynorphin levels are reduced in the epileptogenic focus, while KOR are mostly maintained. Altogether, these results suggest that delivering Dy in affected regions could be of therapeutic value.

The aim of the present study was to replenish the exhausted reservoirs of dynorphins in neurons of the epileptogenic focus. Dynorphin, stored in vesicles can be released and activate KOR to suppress seizures. The paper is well written but more explanations could be directly put in the figures to help reading;

Authors show nice and documented results in terms of prevention of seizures evidently demonstrated by this gene delivery strategy. The use of two different well-recognized animal models in mouse and rat is convincing in the short term. However there are several important concerns that should be addressed to consider the paper for publication

Major comments

1) long-term results

Results on spatial learning and memory are not convincing in the long-term (5,5 months) (Figure 2). This is puzzling. This result and the decreased long term production shown in Figure 3G argue for a decreased expression / efficacy in the long-term. This should be more clearly analyzed and discussed. Escape in the long-term ? Long term production of pDyn should be evaluated.

This is also contrasting with CSF measures (see my comment below discussing results presented page 9)

ad 1)

The overall performance of mice in the Barnes maze is generally decreasing with age. Therefore, both groups, age-matched healthy controls as well as AAV-pDyn treated epileptic animals were analyzed in parallel. The data show comparable performance of age matched naive and pDyn treated mice. The poor performance of aged mice is most probably due to impairments in vision developing over time in C57BL/6N mice. The time-interval chosen is the latest where C57BL/6N mice are able to learn the task, and AAV-pDyn treated epileptic animals keep up with naive mice. This is now also discussed in the MS (see page 6 line 30 onward). Three months after treatment the data are very similar to the ones observed at the 1 month time-interval.

Dyn levels were measured at 1.5 and 6 months after treatment with AAV-pDyn. These data are depicted in Fig 3 G and H. In tissue samples of older mice we see an overall reduction of dynorphin in the hippocampal tissue as compared to the younger group (Fig. 3 G). However, dynorphin is still over-expressed in the ipsilateral hippocampus of AAV-pDyn treated mice. This is also reflected by a slightly increased level of dynorphin in the CSF of AAV-pDyn treated mice at the 6 months after treatment interval (Fig. 3 H). In fact AAV-pDyn treated

animals show a higher level of Dyn in the CSF at 6 months as compared to younger animals. Therefore, we think that the lastingness of the overexpression is well documented. As stated in the manuscript (page 9, line 22), a transient overexpression which then reduces to a lasting steady level is absolutely normal for AAV-vectors.

2) Safety of the approach in naive mice must be addressed more precisely (particularly histology) and in the long-term.

Overexpressing Dyn, even in a restricted portion of the brain could have consequences. Evaluation was made up to 10 weeks after vector administration. This is not sufficient. Long term study is needed.

Cognitive consequences should be evaluated in bilaterally injected animals to increase the sensitivity of the test. Histology should be much more detailed with long-term (at least 6 months) time points and detailed analysis: neuronal aspect (apoptosis, neuronal markers), neuronal count, inflammation (microglia, astrocytes) with high magnification images. Comparison to non-injected animals or contralateral hemisphere should be made. Readouts of Dyn action should be analysed: KOR signaling in particular

ad 2)

We agree that additional safety tests are needed before AAV-pDyn can go into clinical trials. However, such extensive safety testing will be done on the vector produced under GMP conditions and in a GLP-certified laboratory according to the requirements of the regulatory authorities. In the present manuscript, the first "proof of concept" presentation of the feasibility of an AAV-based gene therapy using dynorphin we address a number of relevant safety issues to exclude overt toxicity, inflammation, impairments of behavioral performance, etc., comparing the injected with the contralateral side (as asked for). The extensive analysis the reviewer asks for is absolutely valid before moving the vector into the clinical study and will be designed in collaboration with regulatory boards, but far beyond the scope of our present study. We now indicate the preliminary character of the safety study. Moreover, we deleted the term long-term from the safety study (page..., line...)

3) This release of the dynorphin is called by authors "on demand". This term leads to misinterpretation and should be changed.

Transgenic Dyn is properly matured and behaves like endogenous counterpart and delivered following stimulation. However the term on demand is confusing and not appropriate. The production of the transgenic dyn is not controlled. Overexpression of the pDyn is permanent. Only the release of mature Dyn vesicles is upon stimulation. This notion should be more clearly explained and it is the opinion of the reviewer that the term « on-demand » is confusing, could be misinterpreted, and should not be used particularly in the title. Since one issue with gene therapy is that gene expression is not regulated and transduced gene is permanently expressed, this is of importance.

ad 3)

The reviewer correctly notes that the AAV vector induces permanent expression of pDyn, which is processed to mature, active dynorphins. By contrast, Dyn peptides are only released upon high-frequency stimulation (like at seizure onset), but not during low frequency signaling. And actually, the seizure onset is the situation of demand. Therefore, we consider "release on demand" to correctly capture this process. However, to avoid misinterpretation, we removed the term from the title and introduce it with a clear definition (page 3 line 18)

Even the release of the mature form stored in vesicles seems to rely on stimulation, transgenic Dynorphin is produced constitutively under the CAG promoter which raises questions:

What is the level of production of transgenic Dyn compared to endogenous protein. Total Dyn protein should be quantified.

The amount of transgenic dynorphin in relation to endogenous can be judged from Fig. 3G. In the dorsal hippocampus, the mean protein level of dynorphin is about twice the content of the contralateral side or of either side in GFP-treated animals.

Does it accumulate in cells ? in what compartment ? (high quality histology should be performed to document that) How is it eliminated ?

Authors should investigate the mechanism of action, signal that target correct maturation of pDyn and down signalling mechanism on KOR : show readout of KOR signaling

It is generally accepted that neuropeptides are stored in large dense core vesicles, which are transported into the axon. This is also what we observed by immunohistochemistry for the vector-derived pDyn. We find accumulation of the peptide in axonal, but not in dendritic compartments. This is shown in supplemental figure 3. For this experiment an antibody also detecting precursor peptides was used. Therefore, we can conclude that there is no accumulation or deposits in somata. Elimination of released neuropeptides occurs through degradation by peptidases. As the peptide produced from the vector is identical to the natural one, it is unlikely, that another process would be involved.

Regarding the mechanism of action: In Fig. 1 F we show the effect of a selective kappa opioid receptor antagonist on the seizure suppression by AAV-pDyn. Blocking kappa opioid receptors leads to a re-occurrence of HPDs and seizures to a level comparable to pre-AAV treatment. This provides solid evidence, that the effect is mediated by activation of kappa opioid receptors. KOR activation leads to activation of inhibitory G-proteins. However, inhibition of cyclase or recruitment of beta-arrestin (which also occurs upon binding of Dyn) is not considered as important as the direct interaction of the beta-gamma subunit with voltage-gated calcium channels (presynaptically) or potassium channels (postsynaptically) this was discussed (page 18, line 6 onwards) and is also indicated in Fig. 6 for illustration. Regarding the correct maturation of pDyn: As stated in the manuscript, the antibody used for the ELISA (data shown in Fig. 3) is highly specific for fully matured dynorphin B. It detects Leumorphin, a larger, yet already active form of Dyn B, but only to 7 % and larger fragments not at all. As we detected DynB release in prodynorphin knockout animals after injection of AAV-pDyn with this antibody, it can be assumed that AAV vector-derived pDyn is indeed matured to functionally active dynorphins. Which proportion of pDyn is processed cannot be concluded. This is now stated in the manuscript (page 11, line 24).

Since Dynorphins have important functions in the brain, particularly involving reward processing, mood control and the development of addiction, these issues are of major importance in the perspective of a potential therapeutic use.

We fully agree that dynorphins play an important role in a number of processes. Exactly this reasoning led us to choose an AAV capsids (serotype 1) that barely spreads beyond the site of injection (see Fig. 4A). In addition, our data show that CSF levels of dynorphins stay very low, also several months after treatment (Fig. 3H), suggesting little to no influence on other mechanisms. This is further supported by the lack of alterations in behavior observed in naive mice treated with AAV-pDyn.

Specific comments

Injection protocole :

It would help adding a scheme showing the technics and implanted material. This would help understanding the following Methods : For experiments with subsequent AAV administration, a guide cannula was implanted, which was attached to the hippocampal depth electrode and targeting the hilus of the dentate gyrus. At the time of viral vector injection, animals were implanted with a guide cannula targeting the hilus of the injected hippocampus and a stimulation electrode in the entorhinal cortex (RC -4.20 mm; ML +3.20 mm; DV -4.90 mm). For

We introduced supplemental figure 5 giving a scheme and showing the material used. The placement of electrodes and microdialysis probes for the release experiments is depicted in Fig. 3.

Figure 1 :

The cassette presented in a single strand cassette and not a self complementary cassette. This should be modified. The serotype 2 UTR are used and the capsid is 1 . This should be added.

The figure and legend was changed according to the reviewers suggestions (Fig. 1A)

Legend of Figure 1 :

It is mentioned that control vectors carry either a truncated, nonfunctional version of the enhanced GFP gene (AAV- Δ GFP), or its functional counterpart (AAV-eGFP; not displayed). Why is eGFP mentioned here ? Where is it used ?

AAV-eGFP was used in the behavioral experiments done on naive animals. All other experiments were conducted using with Δ GFP. The respective figure legends states which control vector eGFP or Δ GFP was used.

Figure 3 : How is measured DynB content in G

Figure 3G: Dyn B is measured by EIA in G, H and K. This is now clearly stated in the figure legend (page 10, line 6)

page 9 :

It is stated that « Dyn expression decreases 6 months after application (Fig 3G) and that the early increase of neuronal Dyn did not lead to increased Dyn levels in the cerebrospinal fluid (CSF) at 1.5 months after AAV-pDyn transduction (Fig. 3H). Interestingly a 2-fold increase of Dyn was observed in the CSF at 6 months after AAV pDyn transduction »

This seems in contradiction With 3G results. Moreover this raises the issue of potentially elevated Dyn level in the CSF. In the context of a supposed limited expression in a restricted region of the brain, how this can be interpreted ?

There is an overall reduction of dynorphin levels in all animals at older age. Still, dynorphin expression in the AAV-pDyn injected hippocampus is increased about 2 fold. As the antibody used in the EIA only detects mature dynorphin B, but not its larger precursors, a reduction of neuronal dynorphin paralleled by increased CSF levels can be explained by increased release, but does not allow conclusions on the intracellular expression level.

Regarding the increased CSF levels: Basal dyn levels in CSF are very low and their 2fold increase still represents non-efficacious concentrations in the pM range (about factor 100 below EC50 in the established GTP γ S assay). Thus, already a slight increase in release from a small area may lead to this increase. Considering a total CSF volume of about 30 μ l in mice, a femtomole of dynorphin yields a 30 pM concentration.

Page 11 :

To investigate whether AAV-pDyn derived peptides Dyn peptides suppress seizures-like activity in human

hippocampus . How is this experimental design and the amount of Dyn delivered are relevant to the AAV-pDyn delivery? (in the micodyalisis exp the levels are 200 pm).

The CSF levels measured are 200 pM, however this does not reflect the concentration of dynorphins in the synaptic cleft, but represents only a small, highly diluted portion which escapes direct degradation. We applied 600 nM concentrations of Dyn A and B into the medium, and the peptides have to penetrate the slice section and reach the receptors. Therefore, the concentration on the receptor is probably around EC50 up to a factor of 10-20 above. Such concentration can be reached in close vicinity to the release site also by the endogenous and/or vector derived dynorphins.

Graphic summary : should integrate overexpressing Dyn condition in healthy cells

Epilepsy is considered a disturbance of the network, with distinct groups of cells being more or less vulnerable to damage. Neurochemical changes like the reduction of pDyn expression are not necessarily a sign of “unhealthiness” or damage of neurons, but are induced by increased Calcium influx. Therefore, this may well reflect adaptations to altered network activity. In naive animals, endogenous dynorphins are sufficient to yield the full effect on seizure threshold. This is indicated by the fact, that treatment with KOR agonists does not further increase seizure threshold.

The treatment aims at pDyn overexpression in all living neurons, irrespective of their health state to compensate for reduced pDyn production. The idea of the graphic summary is to illustrate the treatment concept. The bottom panel represents the situation after dynorphin-based gene therapy in epileptic tissue. Therefore, the bottom panel represents healthy cells adapted to a malfunctioning network.

Sup figure 2: images and histological analysis should be strongly improved.

The purpose of this figure is to illustrate the distribution of vector-derived dynorphin in hippocampal neurons. Labeling in the terminal field of mossy fibres is indicative of expression in granule cells, labeling of ipsi- and contralateral projections to the inner molecular layer suggest transduction of mossy cells. In addition, scattered neurons in the polymorph cell layer are labeled. Moreover, the higher magnification images visualize the varicose structure of labeled filaments, characterizing them as axons. As vector-derived and endogenous dynorphin are identical, this experiment was done in pDyn knockout mice. There is no intention to display any potential morphological or histological alterations due to AAV-pDyn injection. This might be part of an extensive safety study, but beyond the aim of a proof of principle study.

Discussion

It is said in the discussion that « A single dose of 10⁸ AAV vector genomes (gp) per animal was sufficient for long-term suppression of hippocampal HDPs and of generalized seizures. The safety profile of AAV-pDyn is excellent: no signs of toxicity, inflammation or immune reactions were recorded. No adverse effects were observed in healthy mice treated with AAVpDyn, suggesting that unilateral hippocampal overexpression of pDyn does not interfere with emotional control or stress coping abilities, effects attributed to KOR activation in distinct brain regions » This must be moderated with regards to my comments both on the limited tox informations (in particular histology) and on the short term study only performed.

We now moderated the text to state that only preliminary tox information is available so far (page 18, lines 7 onwards).

A comment should be made on other gene therapy approaches for this disease and how AAV-Dyn competes with that.

We included a statement on other gene therapy approaches (page 18, line 14 onward).

The conclusion stating that « considering the high medical need for a definite and long-term solution the "drug on demand" concept has enormous advantages over permanent and systemic drug treatment. » is misleading. The « drug » is the AAV-pDYN that could be potentially injected to the patient. The product would be indeed permanently delivered and produced.

The produced product (pDyn) will only be released to the synaptic cleft to reach its site of action (Kappa receptor) upon high frequency stimulation before the onset of seizures, which is "on demand". We replace "drug" by "release".

This conclusion and the sentence below should be strongly moderated, not to oversell the work. « Timely and spatially restricted peptide release limits the development of tolerance and reduces off-target adverse effects, by taking the brain "offtreatment" in inter-ictal phases. Last but not least, it will be fascinating to see the "peptide on demand" concept being transferred to other targets and disease states, and its realization by focused AAV delivery. »

We moderated the statement, which now reads: "Considering the high medical need for a definite and long-term solution the "release on demand" concept has enormous advantages over permanent and systemic drug treatment. Timely and spatially restricted peptide release from a permanently refilled pool limits the development of tolerance and reduces off-target adverse effects, by taking the brain "off-treatment" in inter-ictal phases whilst guaranteeing availability at the onset of seizures."

Again the interesting result of this work is that not only pDyn is produced locally but it is correctly matured and delivered upon epileptic stimulation leading to beneficial effect.

Reviewer 2

Referee #2 (Comments on Novelty/Model System for Author):

The authors employed appropriate model systems on all levels.

Referee #2 (Remarks for Author):

The authors present a range of studies that establish the anti-seizure efficacy and safety of viral vector based preprodynorphin transduction and expression. The studies were well designed in order to address many of the issues inherent to anti-seizure gene therapy. Essentially the diverse findings provided a convincing "proof of principle".

The one minor comment regards the quality of panels A thru C in figure 3. The NeuN immunofluorescence quality is quite low and there are clearly pDYN positive cells that do not co-localize with NeuN. Do the authors have a better example?

The distribution of NeuN in the nucleus is not uniform. This produces a grainy impression of the labeling. Moreover, the images are taken from the granule cell layer of 40 μm sections. In this densely packed layer, cells in different focal planes are labeled. Considering that pDYN is distributed in a different cell compartment than NeuN, it is normal, that mismatches occur, i.e. if a cell on the surface of the section is cut in a way that the nucleus is not present. Therefore, as additional control, we performed co-labeling for pDyn with GFAP, a marker for astrocytes, in parallel. As can be seen from Fig 4 F, the majority of labelled cells sits within the granule cell layer, but does not colocalize with GFAP. Taking both labelings together, we

conclude that a vast majority of transduced cells are actually neurons. This is further supported by the labeling shown in suppl. Fig. 3, displaying dynorphin in the mossy fibres and projections of the mossy cells. Individual axons are labeled in the molecular layer of the dentate gyrus.

Reviewer 3:

Referee #3 (Comments on Novelty/Model System for Author):

This is an interesting paper supporting the idea that viral vector delivery of inhibitory neuropeptides to an epileptic focus could be an effective gene therapy approach for drug-resistant focal epilepsy. They use appropriate rodent models, however a weakness of this paper in my opinion is that their reduction in seizures and epileptiform activity is reported for a relatively small number of animals and during short recording sessions.

We agree that the number of animals in the EEG recordings is not very high. However, effectiveness of the treatment is also shown in tests of cognitive functions of epileptic animals. This increases the total number of animals per treatment group to over 20.

Referee #3 (Remarks for Author):

Agostinho and colleagues present an interesting paper expanding preclinical data studies that indicate viral vector driven over expression of inhibitory neuropeptides can have therapeutic efficiency in rodent models of temporal lobe epilepsy.

'On demand' release of an anti-seizure peptide has several advantages, namely that it is released only during high frequency neuronal firing. This therefore reduces potential concerns of homeostatic compensation in response to permanently expressed and functional proteins as used in some other gene therapy approaches.

The authors demonstrate that this treatment can also improve memory function in previously epileptic rodents.

Below are my comments on the manuscript presented.

1. Although Fig 5a shows that there is a good spread of transduced cells in the dorsal hippocampus can the author's state what the viral transduction volume is? Is expression restricted just to the dorsal hippocampus or does it also transduce the ventral hippocampus. 10⁸ viral particles seems quite low, many studies report much higher titre 10¹²/10¹³.

Thank you for the comment, indeed 2x10⁹ genomic particles were injected in a total volume of 2 μ L. We apologize for the error. The dose has to be seen in relation to the volume of the relatively small volume of the dorsal hippocampus. The ventral hippocampus was mainly untransduced.

2. How long does it take for virally transduced cells to make the protein of interest? Many groups wait 1-3 weeks after AAV transduction to detect sufficient expression of the transgene protein. However in Fig 1B there appears a significant reduction in epileptiform activity and seizures within 1-2 days.

We used self-complementary (sc)AAV vectors, which express markedly faster than single-stranded AAVs. scAAVs express the protein of interest within 1-2 days after injection. We see the full effect after about a week, which fits to the time-course expected for scAAV.

3. The recording period of EEG is very short. 2hrs for HPD analysis and 48hrs for detecting seizures. Seizure frequency in particular tends to be non-normally distributed. Seizures cluster therefore in a 48hr window seizure frequency could be skewed. A longer baseline record period (7-14 days) and longer post treatment recording sessions (preferably continuous) would be more convincing.

We agree that longer periods of recording would help to reduce the influence of non-normally distributed seizures. However, we had to compromise on this, because we need multiple channel recording to identify HPD (unilateral events) and generalized (events also seen over the motor cortices) seizures. The Neurologger provides 4 channels in a wireless system, but recording time is restricted by battery life and data storage capacity. Multi-channel telemetric systems are too large and heavy for the use in mice. And wired systems providing 24/7 measurement frequently are not stable enough on the thin bone of the mouse skull. To reduce variability, we recorded the 2 hours for HPD analysis always between 2 and 4 pm, when seizures are most frequent. Overall, we see high variability in the GFP treated animals, yet statistical analysis reveals a significant improvement in pDyn treated mice due to the strong reduction of seizures and HPDs.

4. Fig 1C shows a blow up of an HPD, can they also provide a blow up of a seizure.

The requested blow up of a seizure was included as supplemental figure 1.

5. Fig 1D - why are the n numbers changing during the course of the experiment - (n=3-7).

The number of animals is decreasing due to the fact, that some animals lost the implant after some time which precluded further recordings. This is now clearly stated in the text. (page 4 line 14)

6. Fig 1F The colour of the preAAV in AAVDyn rats (blue) is a little confusing as the colour blue is used in other panels to denote AAV-GFP.

We changed the color of pre-AAV the columns in panel F and G to black for increased clarity.

7. What is the half-life of norBNI? How long would you expect it to have an effect?

norBNI is detectable in the CNS of rodents for several days, and fully cleared after one week. Therefore, we allowed one week for washout. 24 hours after application is the commonly used time-interval in studies applying norBNI and generally accepted as displaying full effect, while a delayed onset is frequently reported.

8. If there is a release in suppression of HPDs within 0-2hrs post-injection of norBNI does this indicate that dynorphin is constitutively being released?

The reappearance of HPDs upon blockade of the kappa opioid receptor indicates that HPDs are permanently suppressed by dynorphin. However, this is not depending on a constitutive release of dynorphin, but is most likely achieved by short periods of release at each onset of seizures. The stimulation dependence of the release and the absence of constitutive release was shown in the microdialysis experiments (Fig 3).

9. Fig 1G - Control AAV-GFP data is missing from this figure.

Indeed, a GFP group is missing in these experiments. Unfortunately, the drop-out rate due to death during initial seizures or loss of implant (rats rip of the implant, partially together with parts of the skull) is very high and the first group of 16 rats yielded only the 4 rats, which were treated with AAV-pDyn after stable expression of epileptiform activity. We are aware that this shortcoming. Due to ongoing limitation in our animal house, we could only run a

second round of 10 animals, which resulted in a 100 % loss of animals. However, the development and stability of seizures (and the high drop-out) in this model is well described and we added additional pre-recording data before AAV-pDyn application to compensate for this shortcoming. Irrespective of the lack of GFP animals, we consider these data informative as they, together with the data from human tissue, support the findings in our main model.

10. Is the dose of kainic acid correct? Pg 18 line 24, '50nl of 1nM KA', most people use 50-100nl of 20mM KA. Also there is a formatting error here, reference 16.

We apologize for the mistake, indeed it is 1nmol of KA in a volume of 50 nL. We now corrected this in the text (page 19, line 30).

11. 4 electrodes were implanted but analysis only performed on the electrode in the ipsilateral hippocampus? With time there is a development of a more dispersed epileptic network, does dynorphin injection prevent the development of abnormal epileptiform discharges in other brain areas?

In fact we used all four electrodes for analysis to distinguish between focal seizures (HPDs, occur only ipsilaterally) and secondary generalizing seizures, which can be detected from all channels. Contralateral depth and cortical surface electrodes were used to analyse generalization of seizures. We did not observe the development of a secondary focus in the mouse model. By contrast, rats displayed seizure onset in distinct brain regions.

12. Does dynorphin overexpression reduce neuropathology? Is there a reduction in sclerosis, granule layer dispersion relative to animals injected with AAV-GFP?

We started treatment after the onset of self-sustained seizures. In the model used, the neuronal damage and granule cell dispersion in the ipsilateral hippocampus is very broad, already before the treatment with AAV-pDyn. Contralaterally, we observed very little neuronal damage in both groups. Probably due to the relatively low number of generalized seizures, the progression of neuronal damage in the contralateral hippocampus is very slow. Therefore, analysis of neuronal numbers did not reveal significant differences.

13. Is the conclusion that the recovery in memory impairment is due to reduced seizure and HPDs alone or also due to improvement or prevention of worsening in neuropathology?

We assume that the main effect is due to the reduction in seizures, especially because we see an actual recovery of lost learning abilities and not only a conservation. In mice the interconnection between the hemispheres is strong, making an influence of the ipsilateral hippocampus on the contralateral highly likely. Moreover, neuronal damage in the contralateral hippocampus is mild to almost absent, rendering neuropathological alterations rather unlikely.

2nd Editorial Decision

29th July 2019

Thank you for the submission of your revised manuscript to EMBO Molecular Medicine and for infinite patience during the review process. As a matter of fact, we are indeed still missing referee #1 report but we agree with you at this point, we really cannot justify delaying the manuscript processing any longer.

We have received the enclosed report from referee 3. Along with referee 2 who was already supportive of publication and after extensively discussing this case with my colleagues, we are pleased to inform you that we will be able to accept your manuscript pending the following final editorial amendments.

REFEREE REPORTS

Referee #3 (Remarks for Author):

The authors have addressed my previous review questions. I'm surprised that the dose of KA injected results in SE. However the paper will be of considerable interest to the field.

2nd Revision - authors' response

1st August 2019

[The Authors have satisfied all of the referee reports].

YOU MUST COMPLETE ALL CELLS WITH A PINK BACKGROUND ↓
PLEASE NOTE THAT THIS CHECKLIST WILL BE PUBLISHED ALONGSIDE YOUR PAPER

Corresponding Author Name: Regine Heilbronn and Christoph Schwarzer
Journal Submitted to: EMBO Molecular Medicine
Manuscript Number: EMM-2018-09963